# Do Language Models Track Entities Across State Changes?

**Zilu Tang** [* 1]   **Qiao Zhao** [* 1]   **Gabriel Franco** [1]   **Derry Wijaya** [1 2]   **Aaron Mueller** [1]   **Sebastian Schuster** [3]   **Najoung Kim** [1 4]

## Abstract

Entity tracking (ET), the ability to keep track of states, is a fundamental skill that underlies complex reasoning. An increasing amount of work investigates how transformer language models (LMs) solve entity binding *without* state changes. However, there is limited understanding of how non-toy LMs address ET problems of realistic difficulties expressed in natural language. To this end, we investigate the mechanisms underlying ET in more complex scenarios featuring multiple state-changing operations. We find that LMs do not incrementally track world states across tokens or query-relevant states across layers, but simply aggregate relevant information in parallel at the last token when the query becomes evident. We further investigate mechanisms of individual operations (`PUT`, `REMOVE`, `MOVE`) to characterize this non-incremental ET mechanism. Surprisingly, LMs implement the `REMOVE` operation with a fragile global suppression tag; this global removal mechanism predicts various failure modes that we confirm behaviorally. We provide a mechanistic solution of nullifying this tag to partially address this issue. Overall, our findings reveal that LMs solve a fundamentally sequential task using a non-sequential strategy. More broadly, our work illustrates how behavioral and mechanistic analyses can fruitfully interact. Behavioral results inform mechanistic hypotheses, and insights from mechanistic analyses help build stronger behavioral evaluations by predicting failure modes missing from existing evaluations.[1]

---

[*]Equal contribution  [1]Department of Computer Science, Boston University, Boston, USA [2]Department of Data Science, Monash University, Indonesia [3]Faculty of Computer Science, University of Vienna, Austria [4]Department of Linguistics, Boston University, Boston, USA. Correspondence to: Zilu Tang <zilutang@bu.edu>.

*Proceedings of the 43$^{rd}$ International Conference on Machine Learning*, Seoul, South Korea. PMLR 306, 2026. Copyright 2026 by the author(s).

[1]Our code and data can be found at https://github.com/PootieT/entity-tracking-mi.

## 1. Introduction

Entity tracking (ET) is an ability essential for both humans and language models (LMs), recruited in a wide range of task contexts including playing chess (Karvonen, 2024), holding extended conversations across long contexts (Karttunen, 1969; Nieuwland & Van Berkum, 2006b; Kamp et al., 2011), or solving complex reasoning problems (Prakash et al., 2025). One benchmark that evaluates ET capacity is the box dataset of Kim & Schuster (2023) (we refer the readers to this work for a discussion of other targeted ET benchmarks). Evaluations on this benchmark showed that pretrained transformers do exhibit non-trivial (but still limited) ET capacity involving multiple state changes, and that code training significantly improves ET (Kim et al., 2024).

In general, there have been two lines of work that study ET in LMs. One direction, exemplified by Merrill et al. (2024) and Li et al. (2025), studies tracking at the limit by training toy models on ET problems expressed in simplified synthetic languages. Such studies characterize fundamental requirements on the number of layers or tokens needed by transformers to solve ET with specific numbers of state changes. Additionally, Merrill & Sabharwal (2024); Li et al. (2024b); Zhang et al. (2025); Bavandpour et al. (2025) find that chain-of-thought (CoT) can alleviate the layer constraint by offloading computation to the context length. Another line of work (Feng & Steinhardt, 2023; Feng et al., 2024; Prakash et al., 2024; 2025; Dai et al., 2024) aims to understand the mechanisms behind a prerequisite of entity tracking: binding (i.e., the ability to represent and use static states, not considering state changes). Such studies show that models use binding order IDs (the index at which they appear in the context), detectable in the residual stream, to "tag-and-retrieve" bound entities. Recently, Gur-Arieh et al. (2025) also showed that such IDs are actually a combination of three (and possibly more) types of binding mechanisms that vary across contexts. This latter line of work also considers comparatively more naturalistic settings, investigating pretrained models on states expressed in natural language.

While previous mechanistic studies in the naturalistic setting investigate entity binding such as *"The apple is in Box 1. ... Box 1 contains the ___"*, tracking entities in the real world necessitates understanding of how states *change* over

time. Chess pieces *move* throughout the game, information is *introduced* as discourse unfolds, and variables get *deleted* within programs. Although state changes are instrumental to revealing non-trivial tracking capabilities (Kim & Schuster, 2023), they have not been explored mechanistically in non-toy models. Hence, in this work, we ask how models move beyond static binding and *update* their states (i.e. *"... Remove the peach from Box 1. Put the watch in Box 1. Box 1 contains the ___"* (full example in 2.1).

We summarize our contributions as follows:

- We provide evidence that LMs solve ET not by cumulative world building but through retrieval of target tokens in parallel across the context (Sec. 3).
- We provide a mechanistic analysis of how LMs implement individual state-changing operations such as PUT (adding objects) and REMOVE (undoing a binding operation). We characterize PUT as similar to that of a previously described entity binding circuit (Prakash et al., 2024) and REMOVE as a tag which globally suppresses the token to be predicted (Sec 4.1, 4.2).
- We demonstrate that a better understanding of individual mechanisms allows us to predict novel failure modes, augmenting existing behavioral tests, and partially fix them with interventions (Sec. 4.4).

## 2. Experimental Setup

### 2.1. Dataset

> **Example Data**
>
> The apple is in Box 0, the peach is in Box 1, the clock and the jar is in Box 2, the television is in Box 3, the brain is in Box 4, the book is in Box 5, the pin is in Box 6. Put the watch into Box 1. Remove the jar from Box 2. Move the apple in Box 0 to Box 1. Box 1 contains the
>
> Initial Description    Operations    Query

Example 2.1 shows an instance from the box dataset we use (Kim & Schuster, 2023). The first sentence describes the initial contents of each of the seven boxes in the world; we call this sentence the DESCRIPTION. It is followed by sequences of three types of state-changing OPERATIONS:

- PUT: adds new object(s) that do not exist in the current context into a box (*Put the watch into Box 1*).
- REMOVE: removes existing object(s) from one of the boxes (*Remove the jar from Box 2*).
- MOVE: removes existing object(s) from one box and adds them to another (*Move the apple in Box 0 to Box 1*). This operation can be understood as a chain of MOVE-IN (for *Box 1*) and MOVE-OUT (for *Box 0*).

Then, the model must complete the QUERY with the correct contents. See App. A for details of the dataset.

### 2.2. Models

We study GEMMA-2-2B, CODELLAMA-13B, and LLAMA-3.1-70B in a 0-shot completion setting.[2] The main results we present are on CodeLlama-13B, but we supplement results from other models where relevant. CODELLAMA-13B and GEMMA-2-2B show nontrivial performance on single operations and are less computationally expensive; LLAMA-3.1-70B performs well on multiple operations and is hence used for analyses of multi-operation problems. Behavioral accuracies are reported in App. C, G, and H.

## 3. Incremental vs. Non-incremental Tracking

### 3.1. Do LMs track states incrementally over tokens?

Given the sequential nature of the ET task, our first question is whether LMs incrementally build up world states as descriptions and operations unfold. Specifically, we consider the following hypotheses:

1. **H1**: The model builds world states incrementally as it processes the context from left to right. If this were true, there would exist a fixed-length representation (e.g., the last token's residual stream) that encodes the final states of *all* boxes regardless of the queried box (represents *global* states).
2. **H2**: The model *does not* build world states incrementally, and only pulls and consolidates information relevant to the queried box when the query becomes evident (represents *local* states).

We aim to find evidence for or against H1 and H2 by considering two types of linear probes (Belinkov, 2022; Köhn, 2015; Gupta et al., 2015; Tenney et al., 2019). For **H1**, we train 100 **global** 8-way probes, one for each possible object that can be placed in a box in our dataset. Each probe predicts the box in which a specific object is located, with the 0th class indicating that the object is *not present* in any box. If a global probe can accurately predict the location of each object, this would provide evidence for H1: incrementally built world states that are independent of the queried box. For **H2**, we train 100 **local** binary probes (one for each of the 100 possible objects) testing whether each object is present in the *queried* box or not. If the global probe fails to decode the world state but the local probe succeeds at decoding the state of the current box, this would provide evidence for H2. Additionally, we train a set of **mention** probes to see if the model represents whether an object is mentioned in the context or not. This baseline checks whether basic information that is not specific to the queried box is represented at all in

---

[2]See model and prompt details in App. B and I.

the LM. See App. D for more details about the probes.

The input to all probes is the residual stream of the last token "the"[3] (i.e., the final input token before generating the first object in the queried box) at different layers. We train a set of probes independently for each layer. Since each box contains only up to three objects (out of 100) at a time, the probe labels are highly skewed towards the class *not present*. We therefore also report the *non-trivial* probe accuracy which only considers probing results for the objects that have been mentioned in the context. We provide results for CODELLAMA-13B here; see results for LLAMA-3.1-70B and probe training details in App. D.

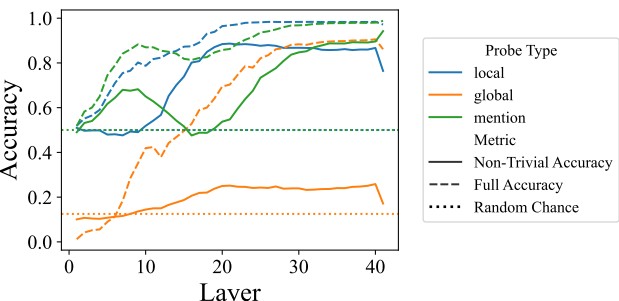

*Figure 1.* Local, global, and mention probe accuracy across layers in CODELLAMA-13B. The low global non-trivial accuracy shows that the model *does not* encode global states in the final token's residual stream, supporting **H2**.

Fig. 1 shows that despite performing consistently above chance ($\approx 0.12$), the global probes' non-trivial accuracy remains lower than 0.3, while local probes' accuracy reaches near 0.9 (chance $\approx 0.5$). The high accuracy of mention probes suggests that some global information is represented, but not organized by box. This provides evidence for **H2**: rather than incrementally updating a set of global states during the processing of the context, **transformer LMs dynamically consolidate only the relevant information when a specific box is queried at the end**.

### 3.2. Do LMs track local states incrementally in layers?

The models do not build global states cumulatively across tokens, but they do encode local states of the queried box at the end. Since the query only becomes evident at the end, incremental tracking specific to the queried box cannot happen across description and operation contexts in an autoregressive model. One possibility is that, if each operation phrase encodes incremental local states of boxes affected by that operation, this information from the last query-relevant operation can be pulled. However, in App. E, we show that incremental local states of operation-relevant boxes are also

---

[3]We also tested representations of all other tokens in the query phrase (e.g., "contains") and the final period token of the last operation, and found "the" to yield the best performance.

*not* decodeable from the operation phrases. Another possibility is incrementally tracking local states across *layers* in the query phrase. We investigate this possibility by asking if prior states—the states of the queried box before state-changing operation(s)—are linearly separable. The more local operations there are, the more prior states an example can have. We outline concrete competing hypothesis and expected results below (also visualized in Fig. 11, App. F):

- **H3**: The model processes (local) state changes sequentially at the last token across layers. If this were true, one would expect to see prior states decodeable earlier in the layer than final states. Probe accuracies for earlier prior states should peak in earlier layers.
- **H4**: The model *does not* process state changes sequentially but in parallel at the last token across layers. If this were true, one would see prior states decodeable with lower accuracy around the same layers as the final states, since prior states are only $n$ operations apart from the final states and partially share labels.

We train sets of 100 binary linear probes (chance $\approx 0.5$) using hidden states at the last token *the* (similar to local state probes), each set of probes targeting different timesteps of local state changes. Each probe detects whether the object is present in the query box at the target timestep. We report CODELLAMA-13B results here and LLAMA-3.1-70B results in App. F.

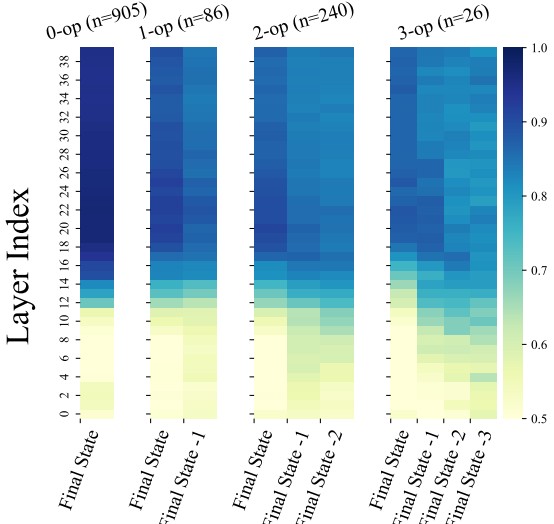

*Figure 2.* Probing for prior states in CODELLAMA-13B reveals that the model also *does not* build states sequentially across layers, supporting **H4**. Each subplot shows a subset of test examples with a fixed number of local operations. Each column within the subplots shows non-trivial accuracy of the final or prior state(s).

As seen in Fig. 2, probe accuracy follows a pattern that resembles **H4** (Fig. 11, right): **LMs do not process state changing operations sequentially in layers.** There is no consistent increase in probe accuracy in earlier layers for

prior state probes; this pattern generalizes across examples with different numbers of local operations. This contrasts with the findings of Li et al. (2025), who finetuned toy models to aggregate state permutations through layers. We suspect the main reason for the difference is that our target entity tokens appear in the contexts, and our operations explicitly refer to the operated objects by name. These explicit entity mentions may disincentivize the models from incrementally representing state changes, and as we will see in the subsequent analyses, this non-incremental processing leads to various failures.

# 4. Mechanisms of State Changing Operations

Our results in Sec. 3 show that LMs do not incrementally track global states over tokens as the descriptions unfold, nor track local states over layers. Rather, they seem to aggregate relevant information from prior contexts in parallel when the query becomes evident. Since state changing operations alter the initial world state, models need a more sophisticated mechanism than, for instance, copying all objects appearing in contexts that mention a particular box ID to succeed on the task. Then, when models do exhibit behavioral success, what are the exact mechanisms behind their implementation of state changing operations? We turn to this question next and investigate PUT in Sec. 4.1 and REMOVE in Sec. 4.2. We treat MOVE semantically as a combination of the other two operations and discuss it in Sec. 4.3.

## 4.1. Put Operation

As illustrated in Example 2.1, PUT introduces a new object and adds it to an existing box. To correctly answer the query, the model must predict the both the PUT object (*watch*) and DESCRIPTION object (*peach*). We hypothesize that PUT follows a similar "look-back" mechanism (Prakash et al., 2025) found for DESCRIPTION (Prakash et al., 2024), thus far only studied with no state-changing operations.

### 4.1.1. CIRCUIT TRACING THROUGH PATH PATCHING

Circuits are end-to-end causal graphs composed of LM components (Olah et al., 2020; Elhage et al., 2021; Conmy et al., 2023b). Typically, circuits are found via counterfactual interventions (Balke & Pearl, 1994; Pearl, 2001) to components (Vig et al., 2020; Geiger et al., 2021) that estimate their degree of influence on the model's behavior. We leverage path patching (Goldowsky-Dill et al., 2023) to sequentially identify four groups of attention heads (named A/B/C/D, following Prakash et al. 2024) responsible for much of the model's behavior (Fig. 3a):

- **Group A**: Active at the last token position in later layers; attends to the target object in the context and increases the target object token logit.

- **Group B**: Active at the last token position in middle layers; attends to the query box ID token in the query phrase and sends target token's positional information (i.e., order ID) to Group A.
- **Group C**: Active at the query box ID in middle layers; attends to the previous box ID token and sends positional information to Group B.
- **Group D**: Active at the previous box ID token in early layers; attends to the entire DESCRIPTION phrase and sends target object position to Group C.

At a high level, we run a forward pass with the original input and obtain the log probability of the target object token ($o$) at the last token position $\log p_{\text{orig}}^o$. We repeat with a counterfactual input and cache the activations. Then, we iteratively patch each attention head's output from clean to patched activations, obtain $\log p_{\text{patch}}^o$, and select the top heads that restore the performance via metric $S^o = \frac{\log p_{\text{patch}}^o - \log p_{\text{orig}}^o}{\log p_{\text{orig}}^o}$. Specifically, we are interested in the similarities between the circuit responsible for DESCRIPTION (what Prakash et al. (2024) characterized) and the circuit for PUT. For this purpose, we focus on examples with one DESCRIPTION object ($o_{\text{desc}}$) and one PUT object ($o_{\text{put}}$). We capture circuits responsible for each object by varying the objective metric $S^o, o \in \{o_{\text{desc}}, o_{\text{put}}\}$ (e.g., $S^{o_{\text{put}}}$ prioritizes heads that promotes the logit of $o_{\text{put}}$, which is the PUT circuit behavior; see App. G for details).

**Similar overall mechanism but different implementation.** In Fig. 3b, we show overlap coefficients ($\frac{|A \cap B|}{\min(|A|,|B|)}$) between the DESCRIPTION and PUT circuits for each group of heads. We find that there are non-trivial overlaps in Groups A and B, while Groups C and D share minimal overlaps. This shows that even though the operations both bind new entities to boxes, they use different sets of heads. This finding extends to GEMMA-2-2B (App. G.2.1).

Model component overlap does not imply functional similarity, or vice versa. To characterize functional similarity between groups of heads in the two circuits, we follow Nikankin et al. (2025) and perform leave-one-out (LOO) analysis, patching heads from DESCRIPTION to PUT (and vice versa), one group at a time. We measure the top-$k$ accuracy: the frequency at which each target object is in the top-$k$ likely tokens, where $k$ is the number of target objects (two here). See App. G.3, G.4 for more details. As shown in Fig. 3c, Group D shows the highest LOO accuracy. This suggests that despite having little overlap in heads and active at different token positions, the heads in Group D in both circuits serve a similar function: to bind objects to box IDs.

### 4.1.2. COMPARING OPERATION SUBSPACES

In addition to group similarity, a major finding from prior work is that models pass positional information (order ID),

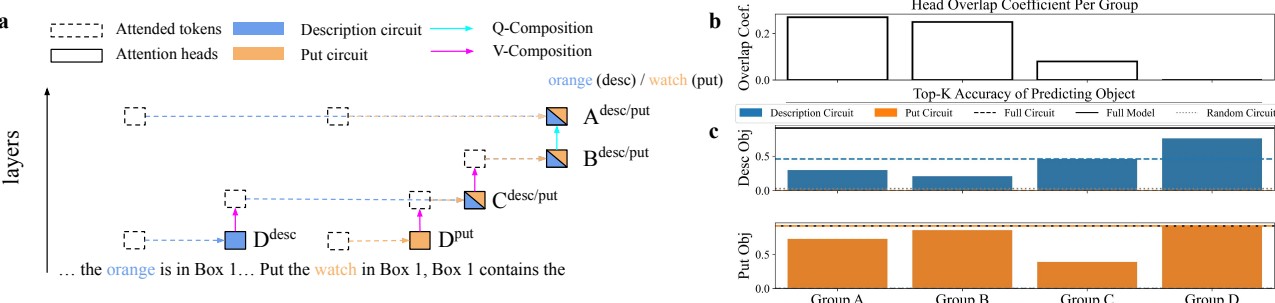

*Figure 3.* `DESCRIPTION` and `PUT` circuits with one `PUT` operation (**a**); their overlap (**b**); and functional similarity across groups using LOO-analysis (**c**) in CODELLAMA-13B. Group A shows the most head overlap while Group D shares the most functional similarity.

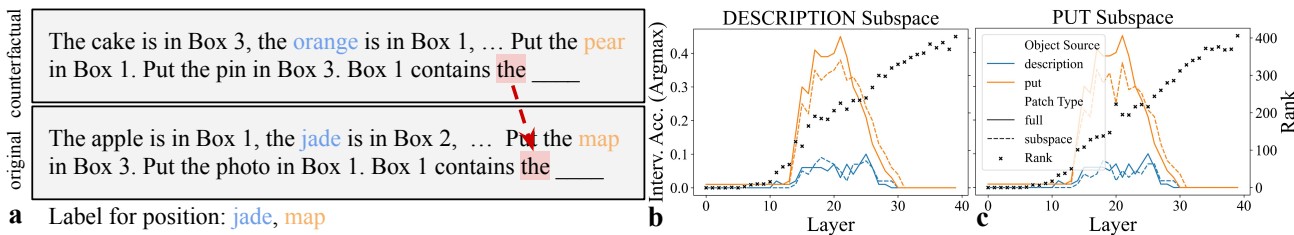

*Figure 4.* Counterfactual design for DCM (**a**); result of subspace activation patching in CODELLAMA-13B (**b,c**). We include two `PUT` phrases, one of which is on the query box. In the counterfactual, all `DESCRIPTION` and `PUT` phrases are shuffled respectively in their groups, and objects mapped to a new set of objects. When patching from the counterfactual to the original sentence at the last token position ("the"), the positional information from Group B heads in the middle layer causes the model to predict **jade** ($o_{\text{desc}}$) and **map** ($o_{\text{put}}$). On the right, we identify the subspaces used in the `DESCRIPTION` (**b**) and `PUT` (**c**) circuits and see that both set of subspaces recover most of the performance for $o_{\text{desc}}$ and $o_{\text{put}}$ alike, indicating similar subspaces used between the two circuits.

not token identity, from Group B to Group A (Prakash et al., 2024; 2025). Using desiderata-based counterfactual masking (DCM; Davies et al., 2023), we design counterfactual examples (Fig. 4a), that under activation patching (Vig et al., 2020; Geiger et al., 2021), reveal whether models transmit positional information used by the `DESCRIPTION` or the `PUT` circuit. In addition to patching the entire residual stream of the last token, we patch subspaces by learning a sparse boolean mask $\mathbf{m} \in [0, 1]^d$ over the PCA-basis of the hidden residual stream space with loss $\mathcal{L} = \text{logit}^o_{\text{patch}} + \lambda \sum \mathbf{m}$, where $d$ is the dimensionality of the hidden states of model, and $\lambda$ is a hyperparameter controlling strength of the mask sparsity. See App. G.5 for details. Crucially, if `DESCRIPTION` and `PUT` were to use different subspaces to send signals, then the subspaces identified by switching the first loss term from $\text{logit}^{o_{\text{desc}}}_{\text{patch}}$ to $\text{logit}^{o_{\text{put}}}_{\text{patch}}$ will reveal different patching behavior.

**PUT and DESCRIPTION circuits use the same subspaces to pass positional information.** In Fig. 4b,c, we show patching results with full activation (solid lines) and with subspaces optimized for either $o_{\text{desc}}$ or $o_{\text{put}}$ (dashed lines). The subspaces faithfully recover the performance of the full activation, as indicated by the close dashed and solid lines. We also show the intervention accuracy for $o_{\text{desc}}$ (blue) and $o_{\text{put}}$ (orange), which measures whether positional information each object is encoded at a given layer. We observe non-

zero intervention accuracy for both solid blue and orange lines in the middle layers, suggesting these layers contain positional information for both objects. This raises the question of whether the two circuits rely on distinct subspaces to represent this information. If the subspaces used for the two circuits are different, we would expect the `DESCRIPTION` subspaces (Fig. 4b) to recover performance for $o_{\text{desc}}$ and not $o_{\text{put}}$, and conversely for `PUT` subspaces (Fig. 4c). Instead, we find that the `DESCRIPTION` subspaces recover performance for $o_{\text{put}}$ just as well as for $o_{\text{desc}}$ (Fig. 4b) and likewise the `PUT` subspaces recover performance for $o_{\text{desc}}$. This suggests that the two circuits rely on functionally equivalent subspaces to encode positional information. Subspace overlap (Fig. 5) supports this finding. We show similar results with GEMMA-2-2B and group A heads in App. G.6 and G.7, respectively. Intervention accuracy is higher for $o_{\text{put}}$ than $o_{\text{desc}}$, consistent with findings from Gur-Arieh et al. (2025) that positional binding is strongest in the beginning and end of the context.

### 4.2. Remove Operation

The `REMOVE` operation removes existing object(s) from the box such that the model should *not* predict these objects (e.g., *jar* in Example 2.1). Analyzing `REMOVE` is fundamentally more challenging than `PUT`, because it hinges on the *absence* of a signal, something that standard interpretability

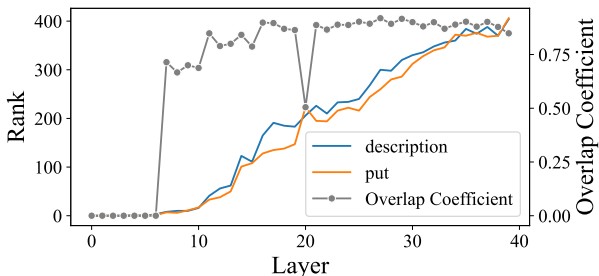

*Figure 5.* Subspace overlap between `DESCRIPTION` and `PUT` circuits in CODELLAMA-13B are high around layers 15-25, where positional information is used.

tools such as activation patching are not designed to capture. Moreover, counterfactuals for `REMOVE` often lead to illegal state changes (e.g., one cannot remove a non-existent object). These limitations make it non-trivial to apply existing techniques on `REMOVE`. Hence, we devise several alternative approaches to be described below.

### 4.2.1. LOGIT AND RANK ANALYSIS

Since removed objects should not be predicted, the mechanism of `REMOVE` (in a model that behaves correctly) must involve logit suppression of the removed object. The degree of such suppression may also depend on whether `REMOVE` is applied to the box being queried. To verify the effect and the scope of the `REMOVE` mechanism, we select a set of sentences with a single `REMOVE` phrase that targets either the query box or an irrelevant box. We obtain the logit and rank for the removed object at the last token ($\text{logit}_{\text{remove}}$) and compare this to the logit/rank of the same object obtained from a context that excludes the `REMOVE` phrase ($\text{logit}_{\text{no-op}}$). We then plot the logit diff$= \text{logit}_{\text{remove}} - \text{logit}_{\text{no-op}}$ (and the corresponding rank diff), binned by whether `REMOVE` is applied to the **query** or **irrelevant** box. We also analyze diffs for the **target** objects (objects in the query box that are not removed) and **all other** objects in the context (any object that is not removed, including target) for comparison. See App. H.2 for examples and more details.

**REMOVE performs *global* removal by indirectly suppressing removed objects.** Perhaps surprisingly, logits for most objects increase when there is a `REMOVE` phrase in the context, even for some of the removed objects (Fig. 6, top). We interpret this as the model simply upweighting most object tokens mentioned in the context. However, the increase for removed objects is significantly smaller than the rest, where query `REMOVE` increases the logit for the removed objects the least (mean close to 0). These leads to the rank of the removed objects increasing relative to all other objects (Fig. 6, bottom), effectively suppressing the prediction of the removed objects. Interestingly, irrelevant `REMOVE` still results in a relative suppression (significant Irrelevant Remove vs. All Other), suggesting a heuristic mechanism

that globally removes objects rather than removing an object from the mentioned box. Similar trends are found with a few-shot setup and in other models (App. H.3). Behaviorally, this *global* removal still yields correct outcomes because of how `REMOVE` is operationalized in the dataset: there exists only one object of the same type across all boxes, so removing an object from the world without considering the box it is in will always be correct. We provide more evidence for global removal in the following sections, and design supplemental evaluation examples that behaviorally disambiguate between global vs. local removal.

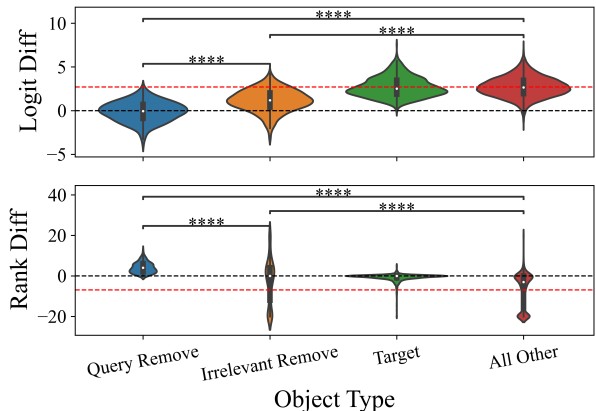

*Figure 6.* Logit and rank diff for `REMOVE` objects in CODELLAMA-13B indicates their rank increase after `REMOVE` phrase and it is observed regardless of whether `REMOVE` targets the queried box. **Black** dotted-line is 0-baseline (i.e. no diff), and **red** dotted-line is the average diff across **all other** objects. We used 2-tail Mann-Whitney test. **\*\*\*\***: $p \leq 0.0001$.

### 4.2.2. TERNARY PROBE FOR REMOVE TAG

The logit and rank analysis shows that models (indirectly) suppress the removed objects' prediction. We hypothesize that models achieve this by tagging objects to be removed with a "remove tag", which causes the model to suppress the predictions later. This is conceptually similar to a binding lookback (Prakash et al., 2025), where a triple of object-box-state is formed in the residual stream. We attempt to localize such a tag by designing a ternary probe to classify the state of each box-object pairs as {`does not exist`, `exists`, `removed`}. We train 700 (each corresponding to an (object, box) pair) ternary probes conditioned on either the object or the box ID token in all description and operation phrases. See App. H.6 for more details, and H.9 for results on LLAMA-3.1-70B. Since most labels are `does not exist`, averaging over all probes leads to trivially high accuracy. Instead, at each conditioning token, we focus on probe accuracy for the box-object pair in the current phrase (**local box-object**, e.g., $(jar, 2)$ for the `REMOVE` phrase in Example 2.1). We additionally plot averaged accuracy across all 100 objects with the current box (**local box**, e.g., $(o, 2), o \in$ all objects) and all 7 boxes with the current object (**local object**, e.g.,

$(jar, b), b \in$ all boxes) to sanity check the presence of object/box information in the representation.

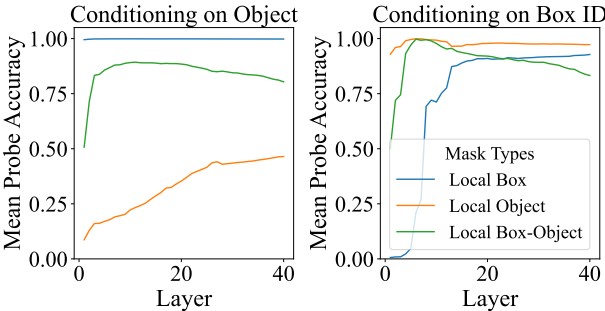

*Figure 7.* High **local box-object** probe accuracy of CODELLAMA-13B conditioned on the **object** and **box ID** tokens suggest both contain "remove tag" signal (around L5-10).

In Fig. 7, we see that **local box-object** accuracy is high at both object and box ID tokens (although higher for box ID), suggesting both could be sources of the remove tag. Upon analyzing probe errors by operation types for the object probe (App. H.7 and Fig. 8, orange lines), we find that most errors stem from the MOVE operation. This is expected as the class label for moved objects is undetermined until the box ID is processed. In App. H.13, we also confirm that remove tags are not accumulated across phrase boundaries as operations unfold and are only found within the specific phrase. Overall, the results show that the remove tag information exists both at the object and the box ID tokens, but it remains to be seen how this information is used in the final consolidation. We return to this question in the intervention experiment (Sec. 4.2.3).

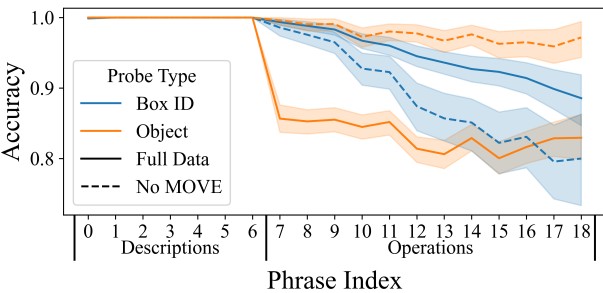

*Figure 8.* Local box-object probe accuracy (CODELLAMA-13B) conditioned on Object and Box ID across phrase index indicates the "tag" signal weakens across context, especially on Box ID.

**Probe accuracy across phrase index.**  Fig. 8 shows the local box-object accuracy across phrase indices at layer 5, where the tag information is the most prominent. The accuracy of the Box ID-conditioned probe linearly decreases with more operations, showing that longer operation chains weaken the tags. Interestingly, if we discard all MOVE examples (since their labels are non-deterministic at the object token), the Object probe is almost perfect, while the Box ID

probe accuracy drops significantly. We perform additional structural analyses of the probe weights in App. H.8.

### 4.2.3. INTERVENTION WITH TERNARY PROBES

Probe classifications provide only correlational evidence. However, one can use probes to intervene on a model's behavior (Ravfogel et al., 2020; 2021). To confirm the causal relevance of the remove tag, we perform intervention experiments on single operation examples. Specifically, we project the hidden states of the token (object and box ID token for respective probes) in the operation phrase into the null space of the probe weights (for details, we refer readers to Ravfogel et al. (2020) and our code) one layer at a time, and roll out model generation. The intuition is that through erasing any signal of the remove/exist tag, we can change the model behavior regarding the object. For an example with one REMOVE operation, we treat **target object** success as cases where the originally removed object is now predicted. We treat **other object(s)** success as cases where other objects in the query box from DESCRIPTION phrase are still predicted. For PUT, we consider **target object** success as the originally put object no longer being predicted. A successful intervention would score high in both metrics (**all objects**). For each operation, we use 100 examples for which the model originally made correct predictions.

**Only the "remove tag" at the object token is causally efficacious.**  To our surprise, intervening at box ID does not affect the removed object, whereas intervening at the object token is successful at earlier layers (Fig. 9, Remove). We also find negative box ID-intervention results by negating the remove tag, or boosting the exist tag, similar to Ravfogel et al. (2021) (App. H.11). This further supports our global removal hypothesis, namely that the removal signal that the models actually *use* are object-level remove tags (rather than box-object-level), resulting in a global removal.

**"Exist tag" confirms previous circuit findings.**  For PUT (Fig. 9, Put), there are a few (early) layers in which the intervention correctly leads to omitting the target object while still keeping the original objects intact (high green and orange lines) at the box ID and, to a lesser extent, at the object token. This suggests both tokens are causally relevant. These signals are likely used by Group D in Fig. 3, as they bind in-phrase objects to the box ID.

### 4.3. Move Operation

MOVE can be semantically understood as PUT and REMOVE. The intervention (Fig. 9) shows similar behaviors between MOVE-IN/PUT, and MOVE-OUT/REMOVE: degeneracy is only predicted for the remove tag and not the exist tag. Thus, our understanding of MOVE is straightforwardly generalizable from the ternary probe results: MOVE is equivalent to

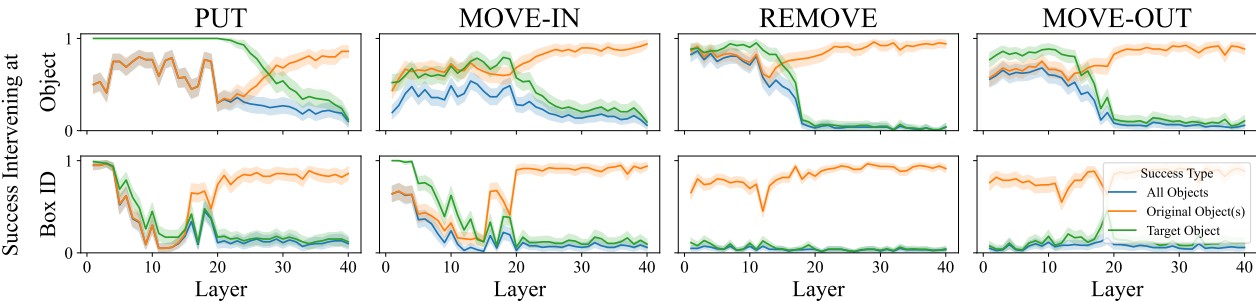

*Figure 9.* Single layer intervention results (CODELLAMA-13B) for nullifying the remove tag (in `REMOVE`/`MOVE-OUT`) and exist tag (in `PUT`/`MOVE-IN`) suggests that the causally relevant remove tag resides in object tokens, while the exist tag can be found in box ID tokens.

adding a remove tag to the object, and an exist tag to the box it is moving into. As shown in Fig. 9 (bottom right), the global removal hypothesis applies to `MOVE-OUT` as well.

### 4.4. Predicting / Fixing Failures of Degenerate **REMOVE**

Our mechanistic analyses reveal limitations of the models that are not evident behaviorally, namely that they employ a global removal mechanism. To this end, we propose three scenarios absent from the original boxes dataset that distinguish global vs. local removal, where our global remove hypothesis predicts behavioral failures:

- **No-op Remove:** *The pill is in Box 0, ... the hat is in Box 3, ... Remove the hat from Box 0. Box 3 contains*
- **Shared-label Objects in Multiple Boxes:** *The pill is in Box 0, ... the jar and the pill are in Box 3, ... Remove the pill from Box 0. Box 3 contains*
- **Re-introducing Removed Object:** *The apple is in Box 0, the peach is in Box 1. ... Remove the peach in Box 1. Put the peach in Box 0. Box 0 contains*

We create 300 sentences for each scenario with a non-empty query box, and greedy decode from CODELLAMA-13B and LLAMA-3.1-70B. We report **recall**, **precision**, and degeneration rate (**DR**, the rate at which the object is globally removed and thus lead to undesirable behavior).

|  | Model | Recall | Prec. | DR(↓) | IS(↑) |
|---|---|---|---|---|---|
| **No-op Remove** | 13B | 0.50 | 0.61 | 0.84 | 0.66 |
|  | 70B | 0.60 | 0.63 | 0.49 | 0.59 |
| **Shared-label Objects** | 13B | 0.69 | 0.85 | 0.56 | 0.73 |
| **in Multiple Boxes** | 70B | 0.75 | 0.85 | 0.27 | 0.74 |
| **Re-introducing** | 13B | 0.61 | 0.85 | 0.62 | 0.24 |
| **Removed Object** | 70B | 0.99 | 0.97 | 0.01 | 0.50 |

*Table 1.* High degeneration rate (**DR**) and the best single-layer intervention success rate (**IS**) confirms predicted failure modes and fixes of our global removal hypothesis.

**Mechanism predicts behavioral failures.** Table 1 shows that all three scenarios result in high DR especially for the 13B model, confirming our behavioral predictions from mechanistic analyses. Aggregating tags is a brittle mech-

anism: it may fail because of diminishing tag strength over long context (Fig. 8), ignored operation order (Re-introduction), and binding failures (No-op, Shared labels).

**Intervention partially fixes failures.** We provide a proof-of-concept study where mechanistic intervention provides a partial solution to the failure. Specifically, for each of the scenarios above, we test whether nullifying the global remove tag recovers the prediction of the object. To this end, we perform single-layer intervention at the object token of the remove phrase for examples where degeneration occurs. We report best single-layer intervention results in the last column (IS) in Table 1. We see high intervention success for the first two cases, suggesting the global remove mechanism is the main culprit for these failure modes. Intervention yields lower success for Re-introduction, and this is likely because Re-introduction requires the models to correctly order multiple operations when consolidating them, and this cannot be fixed by intervention on the removal tags alone.

## 5. Discussion and Future Work

**What is the right way to track entities?** LMs do not build world states incrementally (Sec. 3). Nonetheless, cumulative state tracking remains appealing because it enables constant-time access to any box state. Since ET is a building block for complex reasoning, the architectural limitations identified in the literature is a major bottleneck (Deng et al., 2023; Merrill et al., 2024; Li et al., 2024b; 2025; Bavandpour et al., 2025). This raises interesting questions: what is the most efficient way to enable global tracking at a practical scale (rather than at the limit) within the current architectural limitations? Should tracking be offloaded to external mechanisms? Furthermore, current models tend to only compute goal-relevant variables (Li et al., 2024a), which may contribute further to world states not fully being tracked when queries are unspecified. A possible solution might be to encourage models to latently compute world states (Zelikman et al., 2024).

**Competing mechanisms and CoT.** Our ternary probe accuracies across contexts (Sec. 4.2.2) highlight two interesting trends. First, the better causal signal exists at the box ID

token but decays linearly across context. In contrast, the degenerate signal at the object token, used by LMs causally as the remove tag, remains relatively stable throughout context. This leads to further questions: what are the origins of the preference for the degenerate mechanism? Can we encourage models to prefer more robust mechanisms? This observation about the tag signal decay also provides a mechanistic explanation of why CoT behaviorally improves ET: it decomposes tracking into shorter contexts over which the model has to maintain the tag signals.

**What is the full mechanism for `REMOVE`?** We proposed that `REMOVE` works via the use of remove tags, which are then used in the query phrase. Our initial attempts to pinpoint more specific circuits yielded negative results (App. H.14), suggesting that `REMOVE` differs from previously known mechanisms. Rank difference analyses across models (App. H.3) suggests the presence of box ID-specific mechanisms, but they are too weak to overcome the degenerate global removal mechanism. We leave this to future work. More broadly, our study of `REMOVE` contributes to understanding how LMs *undo* entity binding mechanistically—the inverse of prior work (Wu et al., 2025a). The phenomenon also mirrors the "white bear problem"[4] in psychology, which may offer inspiration for future work.

## 6. Related Work

**Probing world states.** Probing studies use linear classifiers trained on model hidden states to identify signals of interest (Gupta et al., 2015; Belinkov, 2022), which can be used for intervention experiments to establish causality between signal and model behavior (Giulianelli et al., 2018; Ravfogel et al., 2020; 2021; Elazar et al., 2021). Li et al. (2022) show that models trained on synthetic Othello game states represent world states decodeable via probes. In contrast, Mamidanna et al. (2025) find that LMs do not compute intermediate answers when solving multi-step math questions. Work related to latent learning (Lampinen et al., 2025) suggests that pretraining does not encourage learning of information not directly relevant to the task at hand, which may explain the tendency of our models to aggregate relevant context to compute the queried box state as needed, rather than tracking full world states incrementally.

**Entity tracking and binding.** Accurately representing entities across context is a key prerequisite for important skills such as understanding and reasoning (Heim, 2002; Nieuwland & Van Berkum, 2006a; Kamp et al., 2010), potentially underlying LMs' struggle to track long-context dependencies (Merrill & Sabharwal, 2023; Bhattamishra et al., 2020; Deletang et al., 2023; Strobl et al., 2024). Kim & Schuster (2023) propose a controlled ET task in the box-and-objects

format motivated by preliminary investigations (Li et al., 2021; Toshniwal et al., 2022; Li et al., 2022), and Kim et al. (2024) furthermore find that code pretraining is key to surfacing this capacity in LMs. Feng & Steinhardt (2023); Dai et al. (2024); Feng et al. (2024) find that LMs typically solve entity binding tasks by creating an order ID, a low-rank signal in the residual stream, between entity groups. This ID is then used to retrieve the answer token through attention and copying (Prakash et al., 2024) ("look-back" mechanism; Prakash et al. 2025). Unlike our work, these studies did not investigate mechanisms involving state-changing operations.

**Mechanistic interpretability.** The development of causal intervention methods and tool boxes opens up new ways of answering "how" and "why" questions in LMs (Meng et al., 2022; Nanda & Bloom, 2022; Wang et al., 2022; Fiotto-Kaufman et al., 2024). Meng et al. (2022); Wang et al. (2022); Prakash et al. (2024; 2025) use activation and path patching with carefully constructed counterfactual sentences to construct and verify causal models that LMs use to solve specific tasks. Circuit discovery methods leverage approximated attribution algorithms to identify a subset of connected nodes and edges within LMs that perform a task well (Conmy et al., 2023a; Nanda, 2023; Hanna et al., 2024; Hsu et al., 2025). Since causal variables may not align with computational units of transformer models (e.g. attention heads; Geiger et al. 2024), many use subspaces to identify communication signals (Prakash et al., 2025; Merullo et al., 2024; Franco & Crovella, 2025; Wu et al., 2025b) within the residual stream. Our work leverages logit analysis and causal patching to identify entity tracking mechanisms.

## 7. Conclusion

Entity tracking is a fundamental skill necessary for many reasoning problems. Our investigation with naturalistic ET with multiple state-changing operations contributes several findings towards understanding transformer LMs. We find models do not track states incrementally across tokens or layers, but rather, dynamically aggregate information in prior contexts at the final query phrase. Mechanistic investigations reveal that `PUT` resembles an entity binding circuit found in prior work, and `REMOVE` uses remove tag information in the object token to suppress the removed objects. However, the fact that the causally relevant remove tag is found at the object token implies that removal is global, which predicts several degenerate behaviors not evident from existing behavioral tests. Based on this prediction, we design new diagnostic examples that surface the predicted degeneracies. Our work illustrates how behavioral and mechanistic analyses can interact productively. Behavioral tests can inform mechanistic hypotheses, and mechanistic analyses can help build stronger behavioral tests by predicting failure modes missing from existing evaluations.

---

[4]The problem when avoiding thinking about the "white bear" makes one think more about it (Wegner & Schneider, 2003)

# Acknowledgements

This work was supported by a gift from MassMutual awarded to NK. SS was supported by the WWTF through the project "Understanding Language in Context" (WWTF Vienna Research Group VRG23-007). We acknowledge that the computational work reported on in this paper was performed on the Shared Computing Cluster which is administered by Boston University's Research Computing Services. We also used the NNsight package and NDIF in our LLAMA-3.1-70B experiments (Fiotto-Kaufman et al., 2024). We thank Yuhan (Geneva) Yang and Angelos Poulis for their contribution and Mark Crovella for his advice in the early stages of the project. We thank Nikhil Prakash, Dana Arad, Aditya Yedetore, and Martin Tutek for their helpful feedback on experiment designs and analysis. We thank all reviewers for their helpful feedback and suggestions.

# Impact Statement

This paper presents work whose goal is to advance interpretability of language models. Since the ET capability investigated in this work are core to many complex reasoning tasks, there could be second-order impacts of this work if it contributes to developing models capable of more powerful reasoning. However, we do not foresee any immediate ethical or societal impacts deriving from the work presented here directly, since the work does not directly concern applied uses.

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

# A. Dataset Details

The datasets we use for different experiments are controlled to have specific feature distributions based on the target research question. A key difference between our datasets and the original dataset of Kim & Schuster (2023) (the base split) is the format of DESCRIPTION and MOVE. In Kim & Schuster (2023), a DESCRIPTION phrase is of the form *Box 1 contains the apple*, and a MOVE phrase is of the form *Move the apple from Box 1 into Box 2*. In our version, a DESCRIPTION phrase is of the form *The apple is in Box 1*, and MOVE is of the form *Move the apple in Box 1 into Box 2*. We adopted this new format to prevent models from using simple inductions to complete the task for DESCRIPTION, and ensures that object tokens comes before box id tokens in all phrases. We detail the characteristics and statistics of the experimental manipulation in Table 2. Because most experiments require filtering on examples where model succeeds (in first predicted token), we oversample examples in datasets to account for model failures. We describe here the what each column means:

**expObj:** Expected number of objects in a box in the initial state. The number of objects follows a Poisson distribution.

**maxOp:** Maximum number of operations in a sequence that can be applied to any boxes.

**maxObj:** Maximum number of objects in a box at any time. Since our prompts end with *Box X contains the*, we cannot evaluate on examples with an empty query box because the model would have had to generate *Box X contains nothing*. We therefore increase **maxObj** and **expObj** for MOVE-OUT investigations because we need the query box to contain at least one object.

**size:** During dataset construction, we sample each operation sequentially with equal probability until the maximum operation length is reached. Each sampled step is considered a separate datapoint: i.e., operation lengths from 1 through $n$ are individually saved during the process. **size** here refers to the number of unique max length operation sequences. Therefore, the total number of datapoints in each dataset is **size** $\times$ # **maxOp** $\times$ # boxes (which is fixed to seven).

**splits:** Whether train/dev/test splits are used. If used, the fraction is always 0.45 / 0.1 / 0.45.

**noInitEmpty:** Ensures no boxes have empty initial states (i.e. being skipped in the description phrase) by clipping object counts to be one object at the minimum. This is to ensure that all examples contain seven description phrases so there is no significant length difference between examples. This also keeps the semantics of the *PUT* operation consistent — *PUT* operation introduces a new object to a *previously mentioned* box, and not to a *new* box.

**AllowedOps:** Possible operations to be sampled from. We use this to generate datasets that contain only a single type of operations.

If we need to subsample data (like in the case of local /global state probes), subsets are uniformly randomly chosen from respective splits. The **size** of the subsets are around 700 (around 5K sentences). There are total of 100 possible objects in the dataset, all of which are tokenized to a single token by models studied in the paper.

| Version | size | maxOp | maxObj | expObj | splits | noInitEmpty | allowedOps | Experiments |
|---|---|---|---|---|---|---|---|---|
| AltForm_default | 5000 | 12 | 3 | 1 | ✓ | ✓ | {PUT, REMOVE, MOVE} | Local global probes (Sec. 3), Prior State Probe (Sec. 3.2, train-subset, test-subset), Ternary Probe (Sec. 4.2.2, train, test-subset) |
| AltForm_1put_moreObj | 50000 | 1 | 4 | 2 | ✓ | ✓ | {PUT} | Put Path Patching (Sec. 4.1) |
| AltForm_1put_1fixObj | 5000 | 1 | 1 | 1 | | ✓ | {PUT} | Ternary Probe Intervention (Sec. 4.2.3, PUT, DESCRIPTION) |
| AltForm_1put_1_put _irrelevant_1fixObj | 5000 | 2 | 1 | 1 | | ✓ | {PUT} | Put DCM (Sec. 4) |
| AltForm_1remove | 7000 | 1 | 3 | 1 | | ✓ | {REMOVE} | Ternary Probe Intervention (Sec. 4.2.3, REMOVE) |
| AltForm_1move_moreObj | 50000 | 1 | 4 | 2 | ✓ | ✓ | {MOVE} | Ternary Probe Intervention (Sec. 4.2.3, MOVE-OUT) |
| AltForm_1move | 5000 | 1 | 3 | 1 | ✓ | ✓ | {MOVE} | Ternary Probe Intervention (Sec. 4.2.3, MOVE-IN) |
| AltForm_remove_invalid | 300 | 1 | 3 | 1 | | | {REMOVE} | Behavioral Failure (Sec. 4.4) |
| AltForm_remove_duplicate | 300 | 1 | 3 | 1 | | | {REMOVE} | Behavioral Failure (Sec. 4.4) |
| AltForm_remove_put_back | 300 | 1 | 3 | 1 | | | {PUT, REMOVE} | Behavioral Failure (Sec. 4.4) |

*Table 2.* Different dataset versions used in different sections of the experiments.

| Model Names | Huggingface Identifiers |
|---|---|
| GEMMA-2-2B | google/gemma-2-2b |
| CODELLAMA-13B | codellama/CodeLlama-13b-hf |
| LLAMA-3.1-70B | meta-llama/Llama-3.1-70B |

*Table 3.* Models names used in main experiments and their respective Huggingface identifiers

## B. Model Details

We list the models we used in our main experiments along with their respective HuggingFace identifiers in Table 3.

## C. Behavioral Results on Multi-Operation Entity Tracking

We present different models' behavioral accuracy on our `AltForm_default` (test split subset) across the number of state changing operations in Table 4. We use greedy decoding and generate until any punctuation marks or mentioning of another box. If there are multiple objects in the query box, models have to predict the exact set of objects in order for their answer to be considered correct (but the order of the objects can vary). The operation count (State Changing Operation Count) only considers the number of operations that target the query box (i.e., the datapoints corresponding to column **1** may contain more than one operations, but only one of them targets the box we query).

We find that LLAMA-3.1-70B with 0-shot is much more reliable at tracking entities across state changing operations than CODELLAMA-13B. CODELLAMA-13B 0-shot (completion), Although CODELLAMA-13B 0-shot (completion) is limited in its ability to perform multi-operation tracking, we chose to study this model and prompt setting for three reasons. First, 0-shot (completion) has the least amount of tokens, which allows us to perform experiments with reasonable compute. Second, in the local vs. global probing experiments (Sec. 3), we see non-trivial performance of local probes trained on CODELLAMA-13B, which suggest that the model does indeed compute states of the queried box. Furthermore, we verify in mechanistic experiments for PUT (Apx.G.1) and REMOVE (Apx.H.1) that it is capable of performing individual operations well (at least for the first prediction token). We also notice that a large portion of the errors are from the model not being able to predict "nothing" with empty query boxes, this is supported by the second row where CODELLAMA-13B's performance doubles after excluding empty query boxes. Since we exclude such examples for most of our experiments, the second row is a more accurate depiction of CODELLAMA-13B's performance. Unless otherwise mentioned, "0-shot" in the paper is "0-shot" (completion) setting. Prompts can be found in Appendix I.

|  |  |  | State Changing Operation Count |  |  |  |  |  |  |
|---|---|---|---|---|---|---|---|---|---|
| **Model** | **Inf. Bits** | **Prompt** | 1 | 2 | 3 | 4 | 5 | 6 | 7 |
| CODELLAMA-13B | 8 | 0-shot (completion) | 5.1 | 23.1 | 5 | 11.6 | 5.6 | 9.6 | 1.4 |
| CODELLAMA-13B (no-empty) | 8 | 0-shot (completion) | 12.6 | 28.5 | 9.7 | 16.0 | 9.2 | 12.9 | 2.4 |
| CODELLAMA-13B | 8 | 0-shot (instruct) | 31.6 | 32.7 | 9.7 | 17.2 | 10.5 | 8.8 | 6.8 |
| CODELLAMA-13B | 8 | 2-shot | 16.1 | 12.3 | 5.9 | 9.3 | 8.4 | 8.8 | 6.9 |
| CODELLAMA-13B | 8 | 2-shot (all Boxes) | 20.4 | 37.6 | 12.3 | 23.3 | 12.6 | 16.7 | 8.3 |
| LLAMA-3.1-70B | 4 | 0-shot (completion) | 28.8 | 56.3 | 26.5 | 43.3 | 29.8 | 49.7 | 26.4 |
| LLAMA-3.1-70B | 4 | 0-shot (instruct) | 75.2 | 42 | 60.3 | 39.2 | 43.1 | 37.9 | 33.3 |
| LLAMA-3.1-70B | 4 | 2-shot | 91.3 | 73.7 | 72 | 65.5 | 69.4 | 67.4 | 57.3 |
| LLAMA-3.1-70B | 4 | 2-shot (all Boxes) | 91.5 | 78.6 | 74.3 | 73.3 | 64 | 76.8 | 55.3 |

*Table 4.* Behavioral Accuracy (%) for CODELLAMA-13B and LLAMA-3.1-70B with different prompt across different number of state changing operations in `AltForm_default` (test subset).

# D. Details of Local and Global State Probing

## D.1. Probe Label Examples

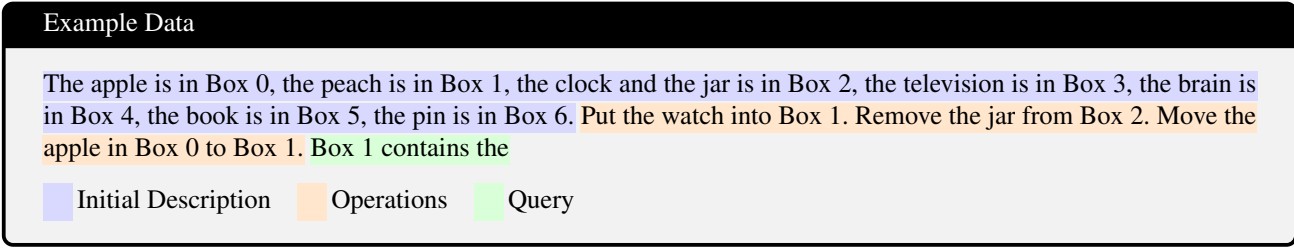

We use the same example (2.1) to illustrate our probe designs. All probes are conditioned on the last token *the*.

- **Global state probes (H1)** need to predict the location of all objects in the final state (i.e. content of all boxes). This is a set of 100 8-way probes, each probe corresponding to an object, and the probe labels corresponding to the location of the object (in Box 0–6 or not present in any box). In this example, 8 probes (corresponding to 8 objects that exist in the world) should predict: *Box 1* (*peach*, *watch*, and *apple* probes); *Box 2* (*clock*); *Box 3* (*television*); *Box 4* (*brain*); *Box 5* (*book*); *Box 6* (*pin*). The remaining 92 probes should predict *not present*.

- **Local state probes (H2)** need to predict the final state for the queried box, which is *Box 1* in this example. This is a set of 100 binary probes, each probe corresponding to an object, and the probe labels corresponding to whether the object exists in the queried box or not. In this example, the *peach*, *watch*, and *apple* probes should predict True, while the remaining 97 probes should predict False.

- **Mention probes** need to predict all objects mentioned in the context. This is a set of 100 binary probes, each probe corresponding to an object, and the probe labels corresponding to whether the object has been mentioned or not. In this example, the *apple*, *peach*, *clock*, *jar*, *television*, *brain*, *book*, *pin*, and *watch* probes should predict True, while the remaining 91 probes should predict False.

## D.2. Probe Training Details

Probes were trained using the `AltForm_default` dataset with the following set of hyperparameters: 64 epochs, batch size of 1024, Adam optimizer, learning rate $1e-3$, betas = $\{0.9, 0.999\}$, with no weight decay. We mitigate class imbalance by setting cross-entropy weights as the inverse of class frequency. Both models use 0-shots (completion) without any prompts.

In Fig. 10, we show **local**, **global**, and **mention** probe accuracy for LLAMA-3.1-70B, which also supports **H2**: LMs do not encode global state consistently in their fixed-length hidden states (low orange solid line).

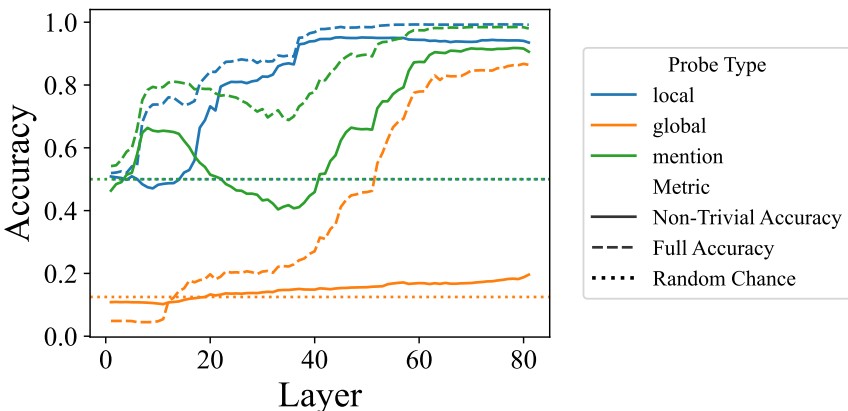

*Figure 10.* **Local**, **global**, and **mention** probes across layers in LLAMA-3.1-70B also reveal that model *does not* encode global state in the final token's residual stream, supporting **H2**.

## E. Incremental State Probing

Sec. 3 provides evidence that LMs' representation at the end of query phrase does not contain global state information, supporting **H2**. Can we trace the local state information to their corresponding phrases? For this question, we train **incremental local** state probes conditioned on the Box ID tokens involved in *each operation phrase* independently, and test whether probes can learn local states cumulated until the current phrase[5]. Similar to **local** probes, we train 100 binary classifiers corresponding to the object in the current phrase (random accuracy is $\approx 0.5$). For instance, Example D.1 would yield the following token-label pairs:

- At token *1* (2nd occurrence), labels are true for *peach* and *watch*, while false for all 98 other probes.

- At token *2* (2nd occurrence), labels are true for *clock*, while false for all 99 other probes.

- At token *0* (2nd occurrence), labels are false for all 100 probes.

- At token *1* (3rd occurrence), labels are true for *peach*, *watch* and *apple*, while false for all 97 other probes.

We tested this with CODELLAMA-13B and LLAMA-3.1-70B. Given results in Fig. 1 showing that the highest probe accuracy is obtained around the final layers, we only trained on the last layer hidden state. Since non-trivial accuracy considers only mentioned objects in the context, which contains 7 boxes up to 3 objects each, the metric can overestimate if probes all predict the class **not present**. To better capture the probe's ability, we additionally report the recall and precision over the **present** class, aggregated across all objects, and found that despite high non-trivial probe accuracy of 88%, the recall is 29% and precision is 13% for CODELLAMA-13B and 85%, 51%, 15% respectively for LLAMA-3.1-70B. Similarly, **local** probe conditioned on the box ID token at the query phrase showed similar behavior. For CODELLAMA-13B, while the non-trivial accuracy is at 77%, recall and precision are respectively (36% and 9%). In contrast, our **local** probe conditioned on *the* (from Fig. 1) has non-trivial accuracy of 89%, recall of 63%, and precision of 57%. This suggests that the **local** state information is aggregated only after the query phrase.

## F. Experiment Details For Prior State Probing

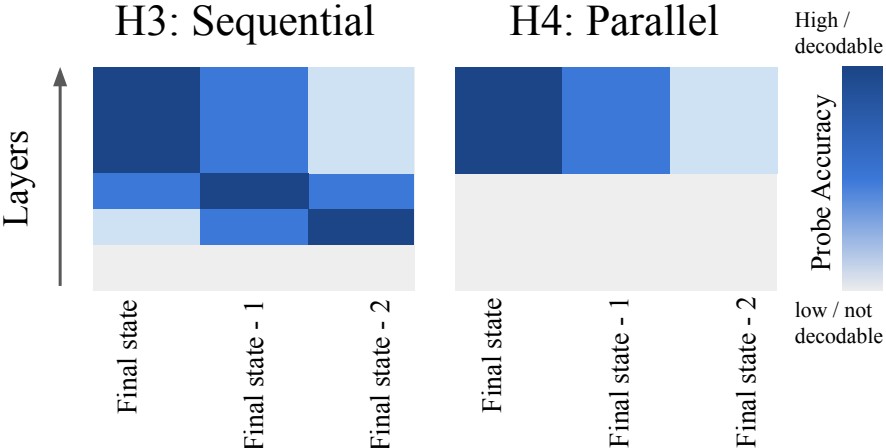

*Figure 11.* Hypothetical probing experiment evidence that would support **H3** (left) and **H4** (right) in example datapoints with two local operations (three different states). If model process states sequentially through layers **H3**, we expect earlier prior states probe to have higher accuracy earlier in the layer. Each column shows probe accuracy of the final or prior state(s).

Probes were trained with the following set of hyperparameters: 64 epochs, batch size of 1024, Adam optimizer, learning rate $1e-3$, betas = $\{0.9, 0.999\}$, with no weight decay. We mitigate class imbalance by setting cross-entropy weights as the inverse of class frequency. Fig. 11 illustrates the hypothetical results supporting either hypothesis (parallel or

---

[5]Note that in our dataset, the Box ID appears after the operator and objects involved in each operation phrase, ensuring the Box ID token has access to the full context of the operation and prior states.

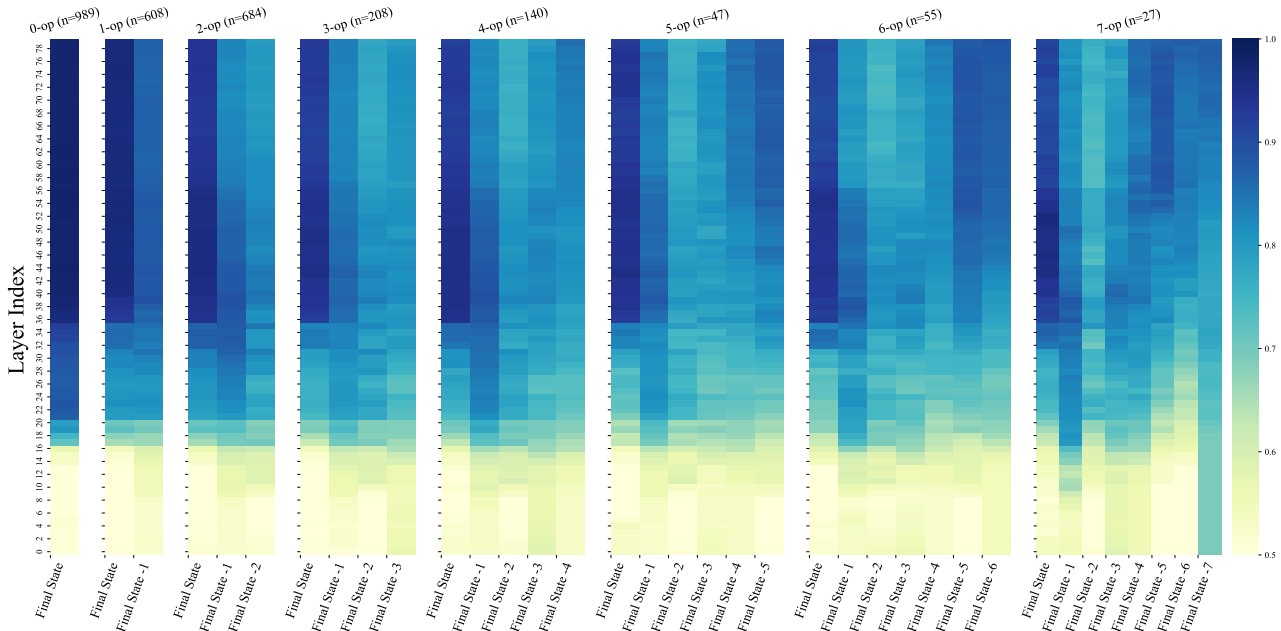

*Figure 12.* Probing for prior states in LLAMA-3.1-70B reveals that the model *does not* build states sequentially across the layer, supporting **H4**. Each subplots shows a subset of test examples having fixed number of local operations. Each column within the subplots shows probe accuracy of the final or prior state(s). As an example, `prior_state=-2` is the next state for `prior_state=-3` and the prior state for `final_state`.

sequential processing across layers). Fig. 12 and Fig. 2 are full probing results for LLAMA-3.1-70B and CODELLAMA-13B, respectively. All results (including Fig. 2 in main text) are aggregated over examples where the model correctly predict the final state through generation (see behavioral evaluation details in App. C) Prompts can be found in Appendix I. All results support **H4**, that state changing operations are processed in parallel across layers.

## G. Put Mechanism Details

### G.1. Behavioral Accuracy

The full generation accuracy (obtained by letting the models generate until the end of sentence token and parsing for predicted objects) for CODELLAMA-13B is around 0.19 (recall=0.66, precision=0.91) in `altForm_1put_moreObj` train subset. Model often predict either all of the objects in the PUT phrase or all of the objects in the DESCRIPTION phrase, but rarely both sets of objects. This is why CODELLAMA-13B has high precision, recall of around half, and low full match accuracy. For patching experiments, we filter datapoints in which the model's first argmax token is one of the target object (we term this logit argmax accuracy), ensuring the model is correct at least at the token in which we are tracing the circuit. Logit argmax accuracy for CODELLAMA-13B 0-shot is 0.91.

In the same dataset, the full generation accuracy for GEMMA-2-2B is 0.01 (recall=0.45, precision=0.63). It behaves generally similarly to CODELLAMA13-B, where it either predicts the contents of the PUT or the DESCRIPTION box (rarely both). In cases where the model has low precision, it either predicts "key" (15% of all cases), a high probability 3-gram completion for "contains the key", or enumerates all objects in the context (15% of all cases). The logit argmax accuracy for GEMMA-2-2B is 0.62.

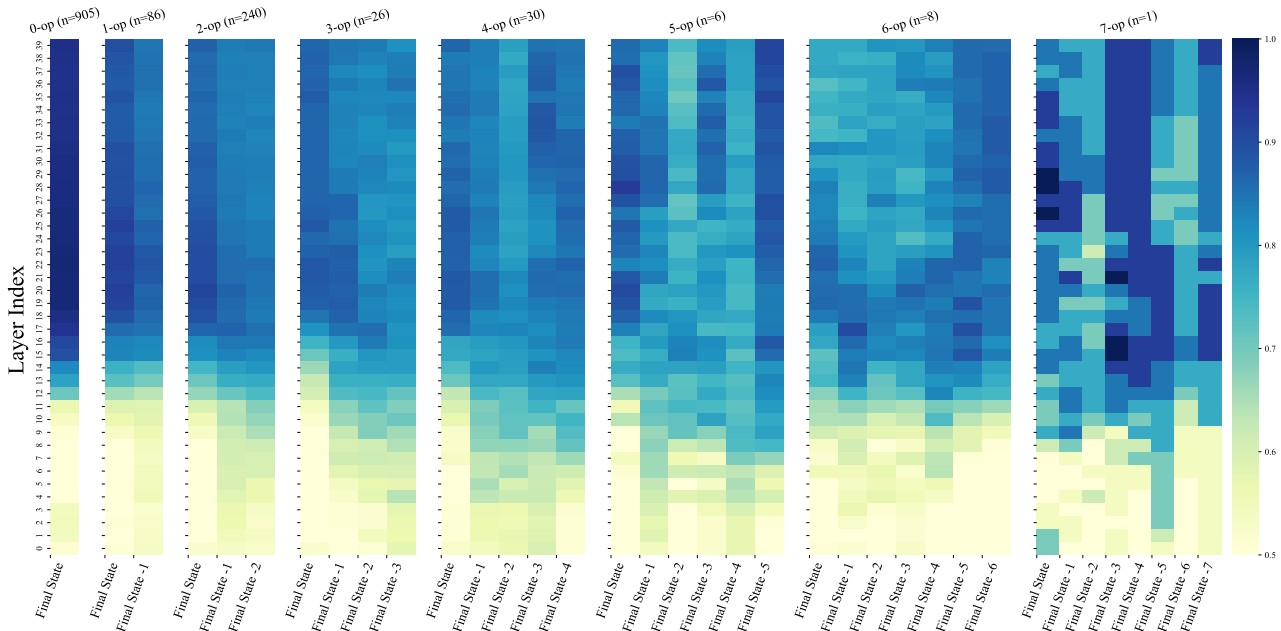

*Figure 13.* Probing for prior states in CODELLAMA-13B reveals that the model also *does not* build states sequentially across the layer, supporting **H4**. Each subplots shows a subset of test examples having fixed number of local operations. Each column within the subplots shows probe accuracy of the final or prior state(s). As an example, `prior_state=-2` is the next state for `prior_state=-3` and the prior state for `final_state`.

## G.2. Path Patching Experiment Details

> **Example Sentence with Counterfactual for Path Patching**
>
> **Original:** The apple is in Box 0, the peach is in Box 1, the clock and the jar is in Box 2, the television is in Box 3, the brain is in Box 4, the book is in Box 5, the pin is in Box 7. Put the watch in Box 1. Remove the jar and the clock from Box 2. Move the apple from Box 0 to Box 1. Box 1 contains
>
> **Counterfactual:** The cup is in Box 0, the pen is in Box 1, the bottle and the paper is in Box 2, the lever is in Box 3, the hat is in Box 4, the phone is in Box 5, the mug is in Box 7. Put the melon in Box 1. Remove the paper and the bottle from Box 2. Move the cup from Box 0 to Box 1. Box 1 contains

Following Prakash et al. (2024), we use path patching to attribute model behavior to a series of four groups of heads across layers responsible for PUT and DESCRIPTION phrases, and for sake of clarity we refer to them as group A/B/C/D. We summarize the functionalities of the groups here, and refer user to details of discovery and functional characterization of the groups to Prakash et al. (2024).

- **Group A**: Active at the last token position around later layers, this group of heads attend to the target object in the context and increase the logit of the object token logit.

- **Group B**: Active at the last token position slightly earlier in layer, this group of heads attend to the query box ID token in the query phrase and inform group A positional information (i.e. Order ID) of the target token through Q-composition.

- **Group C**: Active at the query box ID token in middle layers, this group of heads attend to the previous occurrence of box ID token, and sends positional information to Group B via V-Composition.

- **Group D**: Active at the previous box ID token in early layers, this group of heads attend to the entire DESCRIPTION phrase for the box ID of interest, and supply order ID of the target object to Group C via V-Composition.

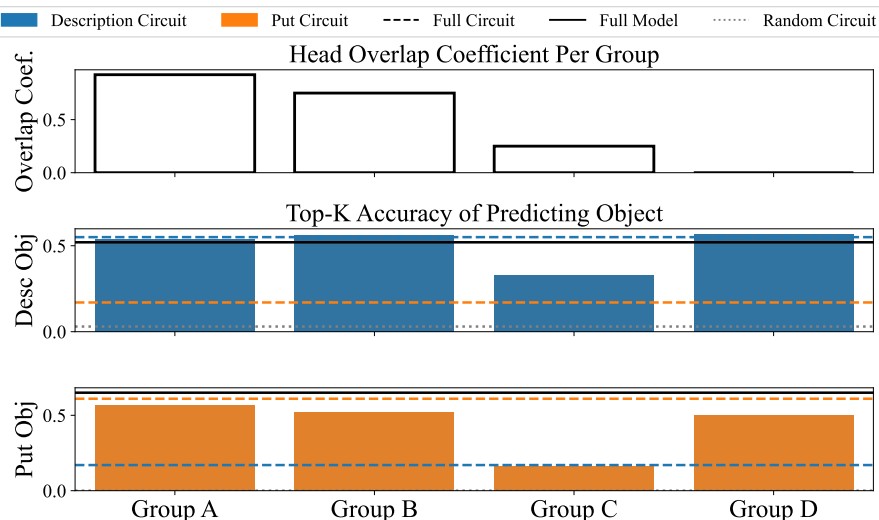

*Figure 14.* GEMMA-2-2B path patching results: attention heads overlap and functional similarity across groups using LOO-analysis. All groups recover a good amount of performance except for group C.

For each datapoint, we generate a counterfactual sentence where each objects are uniquely mapped to a new set of object not in the original context (Example G.2).

For each group, after ranking each heads by the score mentioned in main text, we remove the positive scores and find top head count using the elbow of the negative scores (Satopaa et al., 2011). We then use these as receiver heads to find the next group of heads. We average scores across 200 successful (by logit argmax accuracy) examples to determine head ranking.

For CODELLAMA-13B, the number of heads per group identified through this procedure is (45, 26, 25, 10) for the `DESCRIPTION` circuit and (37, 20, 40, 21) for the `PUT` circuit.

See details on circuit evaluations in Apx. G.3,G.4.

### G.2.1. GEMMA PATH PATCHING CIRCUIT OVERLAP

Since causal patching is computationally intensive for larger models (LLAMA-3.1-70B), we instead verify the generalizability of our findings with a smaller model (GEMMA-2-2B) that adequately handles single operations. GEMMA-2-2B's logit argmax accuracy (0-shot completion) is around 0.71 in `altForm_1put_moreObj`.

In the same way described in Sec. 4.1 and App. G.2, we use path patching to find circuits for `PUT` and `DESCRIPTION` in GEMMA-2-2B (0-shot completion). This resulted in (13,16,12,9) and (14,2,10,7) attention heads per group for the `DESCRIPTION` and `PUT` circuits. During evaluation, we find using heads only from the elbow method does not result in high faithfulness score for the `PUT` circuit, so we increase the attention heads per group one head at a time until we reach 90% circuit faithfulness (i.e. top-k accuracy for $o_{\text{put}}$ of circuit performance is at least 90% of the top-k accuracy of the model). This resulted in (16,4,12,9) attention heads for the `PUT` circuit during evaluation. In Fig. 14, we find high overlaps across `PUT` and `DESCRIPTION` circuits in group A and B, and small overlap in group C and D. For functional similarity, all groups share high similarity except for group C, which seem to differentially promote the prediction of either the $o_{\text{desc}}$ or the $o_{\text{put}}$. Additionally, the full `DESCRIPTION` circuit's performance is better than the full model performance (blue dashed line higher than black solid line, middle plot), which indicates that some heads not included in the `DESCRIPTION` circuit are suppressing the prediction of $o_{\text{desc}}$ (similar to negative name mover heads in the IOI circuit (Wang et al., 2022)).

### G.3. Faithfulness Evaluation of `DESCRIPTION` and `PUT` Circuits

Following prior work in circuit evaluation (Prakash et al., 2024), we use mean ablation to characterize the extent a circuit can reproduce the behavior by keeping only the signals of the heads at certain token positions. We first cache mean activations for each head over 500 random examples in the same dataset. During forward pass, at all tokens except essential tokens (Object, Box ID, Last Token *the*, etc.), we replace attention head outputs with their mean activations for all heads not in the

circuit. We then measure the logit of the target objects at the first prediction token position. If the circuits we identified were essential, the metrics should remain similar.

In our dataset, there could be multiple correct target objects. If we were to evaluate model through autoregressive completion, model can predict "object 1 and object 2" or "object 2 and object 1" and both would be correct. To not penalize order preferences in the patching experiment (since we only measure output of the first token prediction), we calculate top-k accuracy of the objects: frequency at which each object occurs in the top-k likely tokens, where k is the number of correct objects. We evaluate across 100 examples where the model predicts correctly at the first prediction token (i.e. logit argmax accuracy).

### G.4. Leave-one-out Cross Circuit Patching Details

In order to characterize whether the same group of heads in two different circuits perform the same functions, we perform leave-one-out analysis. For instance, when evaluating accuracy for $o_{\text{desc}}$, we start with the DESCRIPTION circuit. To evaluate functional equivalence of group A heads, we exclude attention heads for the DESCRIPTION circuit in group A and include group A heads in the PUT circuit. We then mean ablate all heads not in the circuit at all token positions and measure top-k accuracy of $o_{\text{desc}}$ at the first prediction token. We evaluate across 100 examples where the model predicts correctly at the first prediction token (i.e. logit argmax accuracy).

### G.5. Subspace Patching Details

The assumption of subspace patching is that model components use low-rank signals to communicate with each other (Merullo et al., 2024; Franco & Crovella, 2025). To identify the subspaces, we first cache 500 example activations at the last token position on no-op data. We then use PCA to extract directions as subspace candidates. We use no-op activation cache because it results in better patching performance while using lower-rank subspaces for both no-op and single put data. We then learn a sparse boolean mask $\mathbf{m} \in [0, 1]^d$ over PCA-basis of the hidden residual stream space with loss $\mathcal{L} = \text{logit}^o_{\text{patch}} + \lambda \sum \mathbf{m}$, where $d$ is dimensionality of the hidden states of model, and $\lambda$ is a hyperparameter controlling strength of the mask sparsity. We use 100 separate examples for training and validation sets. For GEMMA-2-2B, the best hyperparameters are $\lambda = 6.0, \text{lr} = 0.02$, and for CODELLAMA-13B, $\lambda = 1.0, \text{lr} = 0.02$.

During circuit evaluation, all subspaces that are not selected by the boolean mask are zero-ed out such that only signals from the desired subspaces are passed through the model.

### G.6. Subspace Patching for GEMMA-2-2B

We also repeat the subspace patching experiments with GEMMA-2-2B and show the results in Fig. 15.

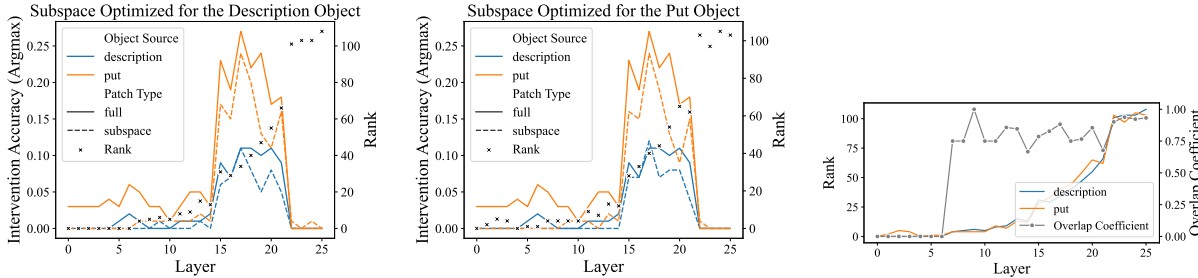

*Figure 15.* Subspace patching results with GEMMA-2-2B confirms that the subspaces used to transmit positional information at the last token layer are highly similar between DESCRIPTION and PUT circuits.

### G.7. Subspace Patching for Token Identity Retrieval (Group A)

Instead of patching with the hypothesis that the residual stream contains positional information about the object, we also use subspace patching to isolate the signals models use to attend to and copy the target object token. Specifically, with the same counterfactual examples as the ones in Fig. 4 (also patching at the last token *the*), we change the labels to orange and pear:

**Counterfactual:**   The cake is in Box 3, the orange is in Box 1, ... Put the pear in Box 1. Put the pin in Box 3. Box 1 contains the ‗‗.

**Original:**   The apple is in Box 1, the jade is in Box 2, ... Put the map in Box 3. Put the photo in Box 1. Box 1 contains the ‗‗.

**Label for object:**   orange and pear

In Fig. 16 we show the results for CODELLAMA-13B. We can see that by optimizing for the `DESCRIPTION` circuit (left), the subspaces recover $o_{\text{desc}}$ just as well as $o_{\text{put}}$, the put objects. In the middle plot, the `PUT` subspaces also recover $o_{\text{desc}}$ equally well. Finally, the right plot shows high subspace overlap around layers 30-40, where the object content information is used. This shows that the subspaces used by group A of both circuits are similar.

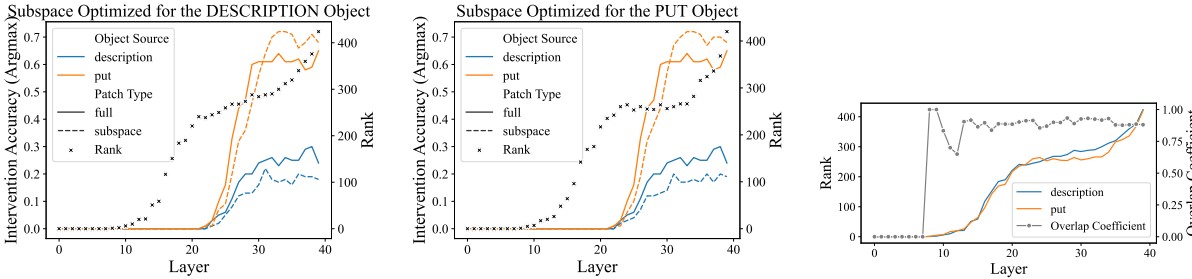

*Figure 16.* Subspace patching results with CODELLAMA-13B confirms that the subspaces used to copy object content information at the last token layer are highly similar between `DESCRIPTION` and `PUT` circuits.

## H. Remove Mechanism Details

### H.1. Behavioral Accuracy

CODELLAMA-13B 0-shot logit argmax accuracy is around 0.84 on the task (`AltForm_1remove`), and the full generation accuracy accuracy is 0.32 (recall=0.83, precision=0.54) on examples where the query box is not empty. In failure cases, the model either predicts the original content of the box (including the removed object) or predicts the left-over content of the box along with other objects adjacent to the queried box. This behavior results in high recall and low precision/accuracy. Since the left-over objects usually have the highest probability of being predicted, the logit argmax accuracy is still reasonable.

### H.2. Rank Diff Details

In the rank diff experiment, we use the following pairs of sentences to isolate the effect of `REMOVE` applied to the query box and an irrelevant box. Note that for the irrelevant `REMOVE`, we switched the contents of the query and irrelevant boxes from respective no-op sentences so we measure the logit diff of the same object across two pairs of sentences.

**No-op (query):**   "The apple and the cake are in Box 0, the banana is in Box 1, the book is in Box 2, ... Box 0 contains ‗‗ ." Measure logit of "apple".

**1-`REMOVE` (query):**   "The apple and the cake are in Box 0, the banana is in Box 1, the book is in Box 2, ... Remove the apple from Box 0. Box 0 contains ‗‗ ." Measure logit of "apple".

**No-op (irrelevant):**   "The banana is in Box 0, the apple and the cake are in Box 1, the book is in Box 2, ... Box 0 contains ‗‗ ." Measure logit of "apple".

**1-`REMOVE` (irrelevant):**   "The banana is in Box 0, the apple and the cake are in Box 1, the book is in Box 2, ... Remove the apple from Box 1. Box 0 contains ‗‗ ." Measure logit of "apple".

In the 1-`REMOVE` (query) example, **other** objects refer to all objects not in *Box 0* in the final state (e.g. *banana*, *book*, ...). The **target** object refers to the *cake*.

In Sec. 4.2.1, we aggregate across 500 query and 500 irrelevant remove examples. We filter examples such that there are two target objects in the queried box in the no-op example so there is always a single remaining target object in the box to be predicted (i.e. the box is not empty). This is to ensure argmax logit accuracy calculation is correct: since we prefix the model with "Box X contains the", the queried box cannot be empty. We clip the rank difference between $[20, -20]$ as changes bigger than that are at the very long tails. In all remove phrases, we remove the first object of the box so that the models, which often generate answers in the same order as they appear in the context, have to change their generations to correctly process the `REMOVE` phrase. Only successful examples where the first prediction token (logit argmax accuracy) is correct is shown in the plot.

In addition to the logit/rank diff analysis in the main text (Fig. 6), we also investigated whether CODELLAMA-13B's logit argmax accuracy correlates with differences in rank or logit. If behavioral failure is caused by the lack of logit suppression of the removed object, we expect high correlation between accuracy and rank/logit diff. We show the results in Fig. 17 (left and right respectively). The logit suppression shows no difference but the rank increase is slightly less among examples where the model predicts correctly. We also do not see a consistent pattern with other models (Sec. H.3), suggesting that behavioral failures in remove is likely caused by factors other than the logit suppression malfunction.

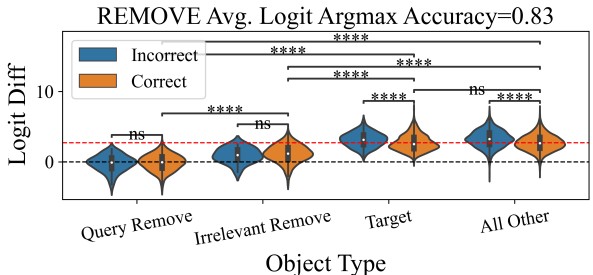 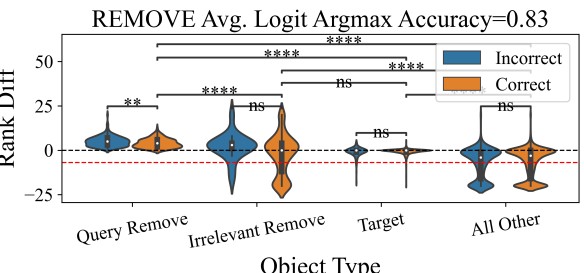

*Figure 17.* Logit (left) and rank (right) diff of different objects before and after adding a `REMOVE` phrase (CODELLAMA-13B). Each examples are also split by model logit argmax correctness. We see similar trend between two metrics, and no significant effect of model correctness on either metric. **Black** dotted lines denote the 0-baseline (i.e. no diff), and **red** dotted lines denote average rank diff across all **other** objects in the correct examples. We used 2-tail Mann-Whitney test for stastical significance. **ns**: $0.05 < p \le 1$, **\***: $0.01 < p \le 0.05$, **\*\***: $0.001 < p \le 0.01$, **\*\*\***: $0.0001 < p \le 0.001$, and **\*\*\*\***: $p \le 0.0001$

### H.3. Remove Rank Diff for Other Models

In this section, we provide the rank difference plot for 1-`REMOVE` case in main text for three families of models (using the same experimental setup as Sec. 4.2.1 and Appendix H.2): LLAMA (Fig. 18), QWEN (Fig. 19), GEMMA (Fig. 20), and MISTRAL (Fig. 21). We report several findings from these results:

**Few-shot prompts change baseline behaviors.** In zero-shot settings (bottom rows), most models show a rank drop for **other** objects (red dotted lines are below black dotted lines). In two-shot settings (top rows), the drops become almost insignificant, and in some cases, turn positive. A potential explanation is that in the zero-shot setup, the removal phrase is helpful in inferring what the task is, thereby helping the model to narrow down the correct set of candidate output tokens (i.e., any object in the context). In the two-shot setup, the in-context demonstrations already sufficiently support task inference, so a similar overall drop in object logits does not occur.

**Consistent rank increase in query and irrelevant remove objects.** In most models, the rank differences for query remove is greater than irrelevant remove, which is greater than other objects in zero- and few-shot cases. The differences are mostly statistically significant. This suggests that there is a logit suppressing mechanism for the removed objects, and the scope of such mechanism affects irrelevantly removed objects as well but to a lesser extent.

**Global remove is observed in non-code pretrained models as well.** Previous work find code pre-training beneficial for LMs' entity tracking performance (Kim & Schuster, 2023; Kim et al., 2024; Prakash et al., 2024). Since the main models we study (i.e. CODELLAMA-13B) is finetuned on code data, it is uncertain whether non-code pretrained models also perform global remove. Among all the models we tested, LLAMA-2-7B, LLAMA-2-13B, and MISTRAL-7B are models not pretrained with code data. In all except LLAMA-2-7B, we see a significant divergence between irrelevant `REMOVE` rank diff

and other object rank diffs. This suggests that the global remove phenomenon is likely not a by-product of pretraining on code data.

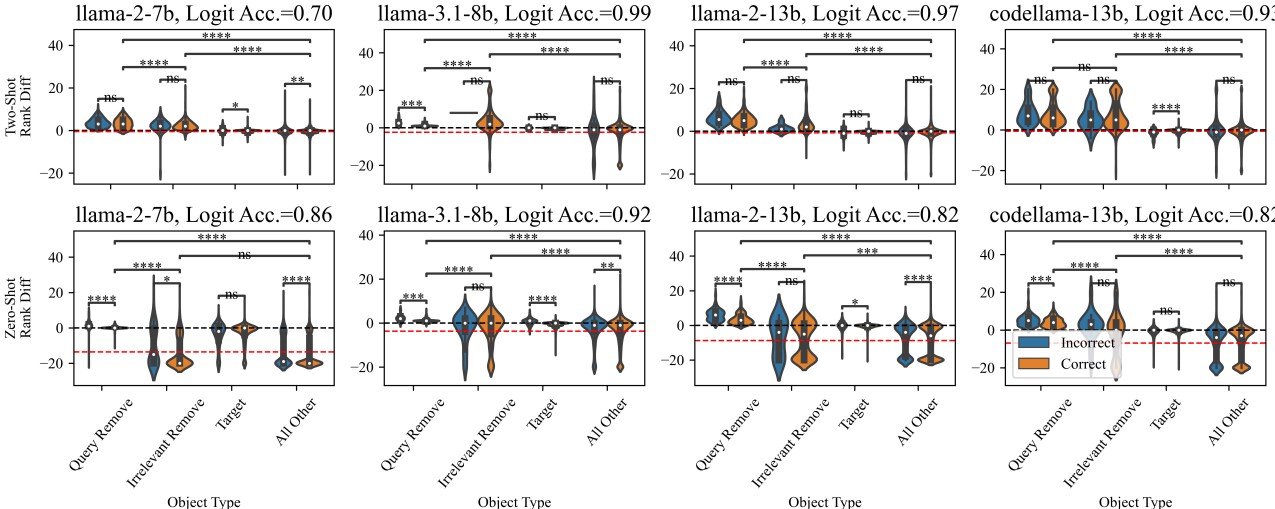

*Figure 18.* Rank Difference of object with 1-REMOVE operation that is applied on either the query box or an irrelevant box for LLAMA models with/without 2-shot prompts. **Black** dotted-line is 0-baseline (i.e. no diff), and **red** dotted-line is average rank diff across all **other** objects in the correct cases. We used 2-tail Mann-Whitney test. **ns**: $0.05 < p \le 1$, **\***: $0.01 < p \le 0.05$, **\*\***: $0.001 < p \le 0.01$, **\*\*\***: $0.0001 < p \le 0.001$, and **\*\*\*\***: $p \le 0.0001$.

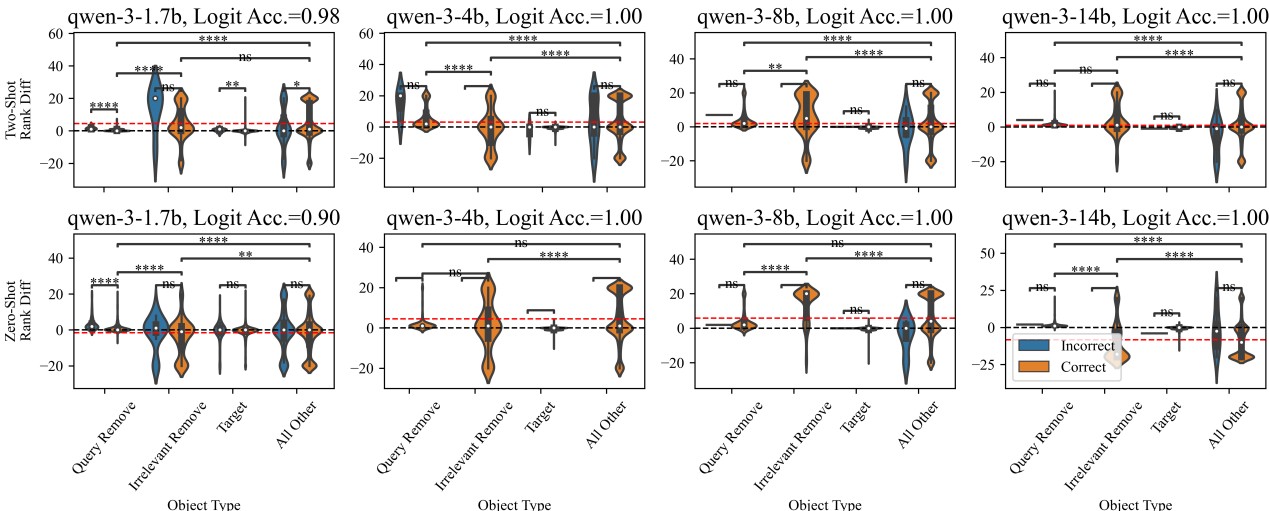

*Figure 19.* Rank difference of object with 1-REMOVE operation that is applied on either the query box or an irrelevant box for QWEN models with/without 2-shot prompts. **Black** dotted-line is 0-baseline (i.e. no diff), and **red** dotted-line is average rank diff across all **other** objects in the correct cases. We used 2-tail Mann-Whitney test. **ns**: $0.05 < p \le 1$, **\***: $0.01 < p \le 0.05$, **\*\***: $0.001 < p \le 0.01$, **\*\*\***: $0.0001 < p \le 0.001$, and **\*\*\*\***: $p \le 0.0001$.

### H.4. Remove Rank Diff by Position

Gur-Arieh et al. (2025) found that the positional IDs that LMs use to bind entities have a diffused and Gaussian-like effect on the entities adjacent to the target entity—the closer the neighboring entities are, the higher the effects. In our logit/rank diff experiment (Fig. 6), we observe that the logit suppression from the REMOVE operation also affects irrelevant boxes. We are curious whether there are position specific effects from where the queried box is within the context (e.g. "box 1" vs. "box 4"), similar to findings from Gur-Arieh et al. (2025). We sampled query and irrelevant remove sentences and keep

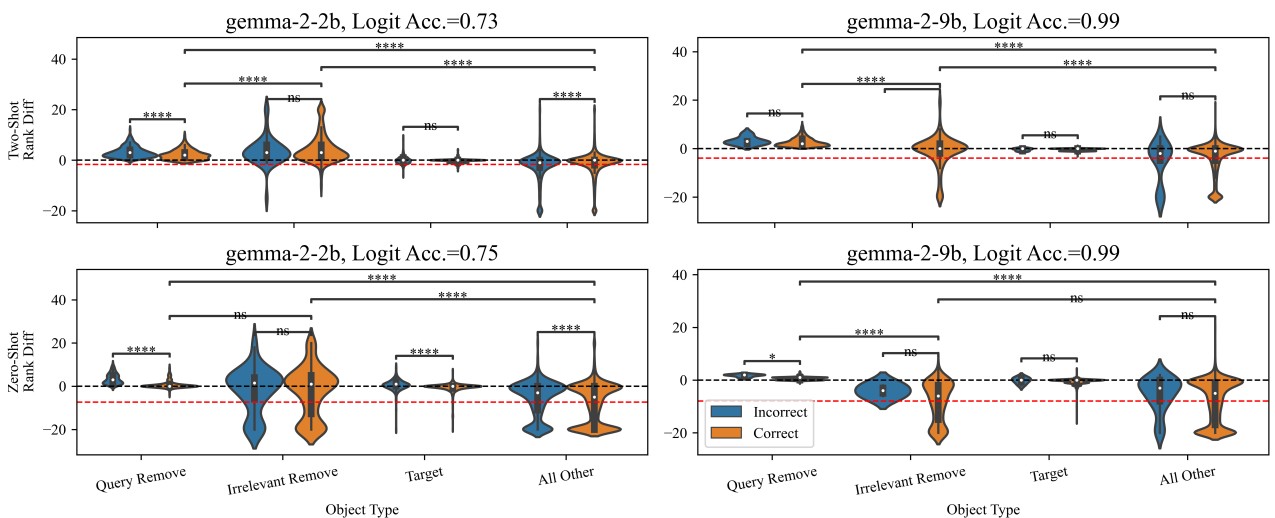

*Figure 20.* Rank difference of object with 1-`REMOVE` operation that is applied on either the query box or an irrelevant box for GEMMA models with/without 2-shot prompts. **Black** dotted-line is 0-baseline (i.e. no diff), and **red** dotted-line is average rank diff across all **other** objects in the correct cases. We used 2-tail Mann-Whitney test. **ns**: $0.05 < p \leq 1$, **\***: $0.01 < p \leq 0.05$, **\*\***: $0.001 < p \leq 0.01$, **\*\*\***: $0.0001 < p \leq 0.001$, and **\*\*\*\***: $p \leq 0.0001$.

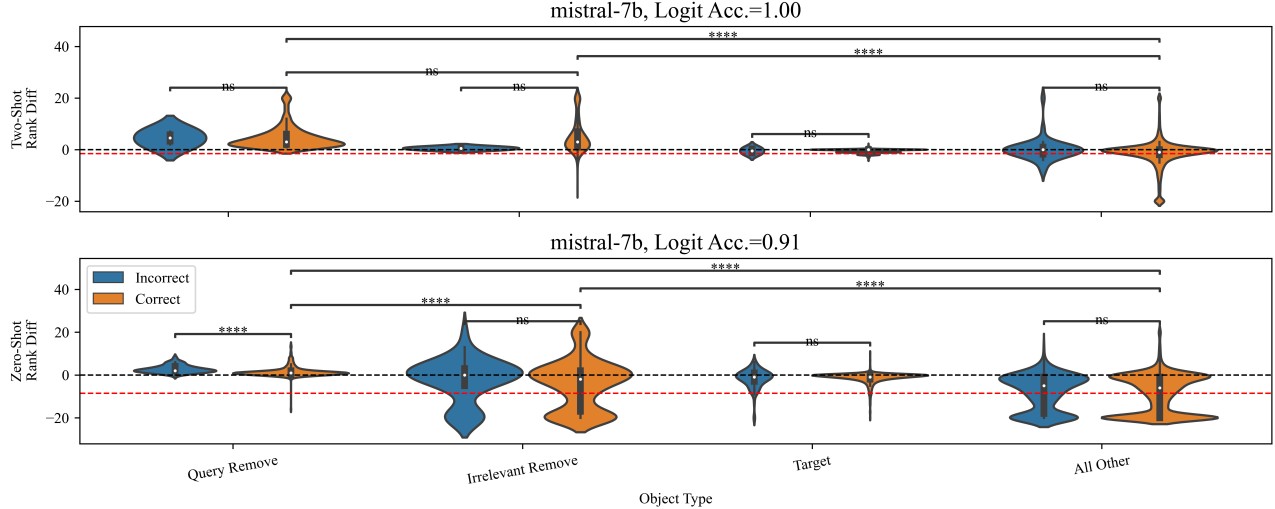

*Figure 21.* Rank difference of object with 1-`REMOVE` operation that is applied on either the query box or an irrelevant box for MISTRAL model with/without 2-shot prompts. **Black** dotted-line is 0-baseline (i.e. no diff), and **red** dotted-line is average rank diff across all **other** objects in the correct cases. We used 2-tail Mann-Whitney test. **ns**: $0.05 < p \leq 1$, **\***: $0.01 < p \leq 0.05$, **\*\***: $0.001 < p \leq 0.01$, **\*\*\***: $0.0001 < p \leq 0.001$, and **\*\*\*\***: $p \leq 0.0001$.

only the behaviorally correct remove examples (by logit argmax accuracy). We sample 10K total sentences for 0-shot and 7.6K total sentences for 2-shot to ensure statistical significance. We plot the 0-shot (Fig. 22) and 2-shot (Fig. 23) results for CODELLAMA-13B, and 2-shot results for GEMMA-2-9B (Fig. 24).

**Remove signal has diffused effect across positions.** In Fig. 22 (right column), we see that the rank diffs for objects in an **irrelevant** REMOVE are visibly higher (and most of the time statistically significant) near the query box than those away from it. The effect is weaker when the query box is towards the middle of the context (i.e. Box 3 or 4; middle rows subplots). Similarly, in the left column, we see the same pattern for the **other** objects. This is similar to what Gur-Arieh et al. (2025) found regarding the diffused positional information used to bind entities, which suggest that the *entity un-binding* process

also leverages some kind of positional binding.

**Few-shot prompts potentially change mechanisms.** Nevertheless, we see much less, if any, diffused positional effects in the 2-shot setup (Fig. 23), suggesting that few-shot learning potentially alters the nature of entity tracking mechanism that the models implement.However, different from CODELLAMA-13B, the diffused effect from **irrelevant** REMOVE is still evident in GEMMA-2-9B 2-shot cases (Fig. 24). This suggest mechanistic effects of few-shot prompting may be different for models.

### H.5. Put Rank Diff

We also perform a similar experiment tracking rank and logit differences of the object introduced by a PUT phrase and provide the results in Fig. 25 (CODELLAMA-13B, 0-shot). Logits for $o_{\text{put}}$ increase regardless of the operant box, and the logit increase for the irrelevant box is smaller than that of a query box. The resulting rank difference is quite similar between query and irrelevant. Similar patterns are observed in the two-shot setting (Fig. 26) and in other models (Fig. 27). This suggests that PUT is also somewhat global. Unlike REMOVE, PUT is introducing new objects that often have not appeared in the context before, this increases the logits (and decrease the ranks) of the PUT objects significantly from the no-op sentences. This effect on irrelevant PUT is also likely caused by model heuristics that does not track states but simply predict any objects (Sec 4.2.1) or enumerate objects from the start (Apx H.10).

### H.6. Ternary Probe Training Details

To isolate signals of "remove-tag", we designed a ternary probe to specify three distinct states of each box-object pair: {does not exist, exists, removed}. This is designed as 7 (boxes) × 100 (objects) linear classifiers that predict three classes given the hidden states of the model at a single layer. Probes were trained with the following hyperparameters: 5 epochs, batch size of 1024, Adam optimizer, learning rate $1e - 4$, betas = $\{0.9, 0.999\}$, with no weight decay. We mitigate class imbalance by setting cross-entropy weights as the inverse of class frequency. We also train on the full training set (AltForm_default) as opposed to the subset (like other probes) to increase the examples in the exists/removed classes. We use 0-shot for both CODELLAMA-13B and LLAMA-3.1-70B.

### H.7. Ternary Probe Error Analysis

We present the ternary probe confusion matrices here for CODELLAMA-13B (Fig. 28, Layer 8) and LLAMA-3.1-70B (Fig. 33, Layer 10), where the probe accuracy is near maximum performance. We see that for the object probes, majority of the confusion comes from the MOVE operation, while this is not the case for Box ID probes. Confusion for MOVE is expected for the object probes given the label of the probe (exists or removed) is not determined until later in the sentence (e.g. in the example *Move the apple in Box 1 to Box 2.*, the label for (box 1, apple) should be removed, but this cannot be determined until at least the token *1*).

### H.8. Probe Structural Analysis

In addition to accuracy, we also analyze the structural properties of our probes. One dependency between the classes is that an object cannot be removed without the object having existed first. This suggests that objects that are removed or exists should be closer to each other than objects that have never existed in the world.

Based on this intuition, we hypothesize that probe weights for removed are more similar to exists than to non-exist, and that the removed and exists weight vectors are largely captured by a low-dimensional subspace spanned by non-exist probes (i.e., projecting removed/exists onto the top PCs of non-exist would yield relatively low reconstruction loss), whereas the converse projection should lose more information. To confirm this, we plot two set of metrics between probes at each layer. First, we plot mean and standard deviation of pairwise **cosine similarity** across all 700 object-box pairs of probes weights between two probe classes. Second, we construct low-rank PCA projection matrix using source class probe weights (that recover 50% of the variance) to reconstruct target class probe weights, and measure **reconstructed weight norm ratio** against the original weight . By symmetric nature of such metric, if the non-exist PCA subspace yields a higher reconstruction norm ratio for removed than the reverse direction, this indicates that removed probe weights are largely contained in (or well-approximated by) a low-dimensional subspace already present in the non-exist weights. Lastly, we plot the number of principal components needed to recovers 50% of the variance for each of the probes.

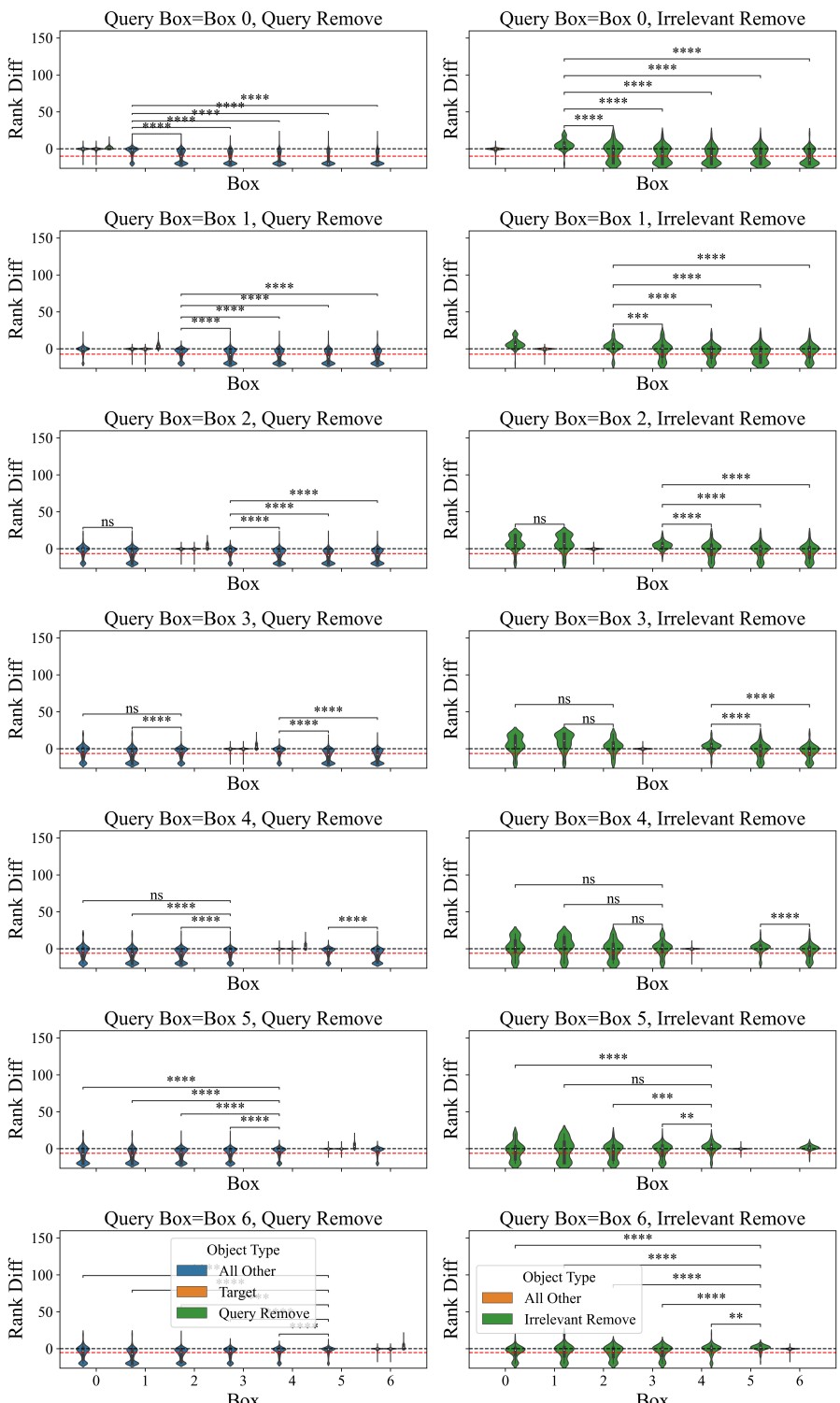

*Figure 22.* REMOVE phrase rank diff split by query box position (CODELLAMA-13B, 0-shot) shows that the effect of rank increase in **irrelevant REMOVE** diffuses across box positions (similarly for **other** objects), but such effect is less pronounced when the query box is in the middle of the context. **Black** dotted-line is 0-baseline (i.e. no rank change), and **red** dotted-line is average rank diff across all **other** objects. We use 1-tail Mann-Whitney test for statistical significance. **ns**: $0.05 < p \le 1$, *: $0.01 < p \le 0.05$, **: $0.001 < p \le 0.01$, ***: $0.0001 < p \le 0.001$, and ****: $p \le 0.0001$

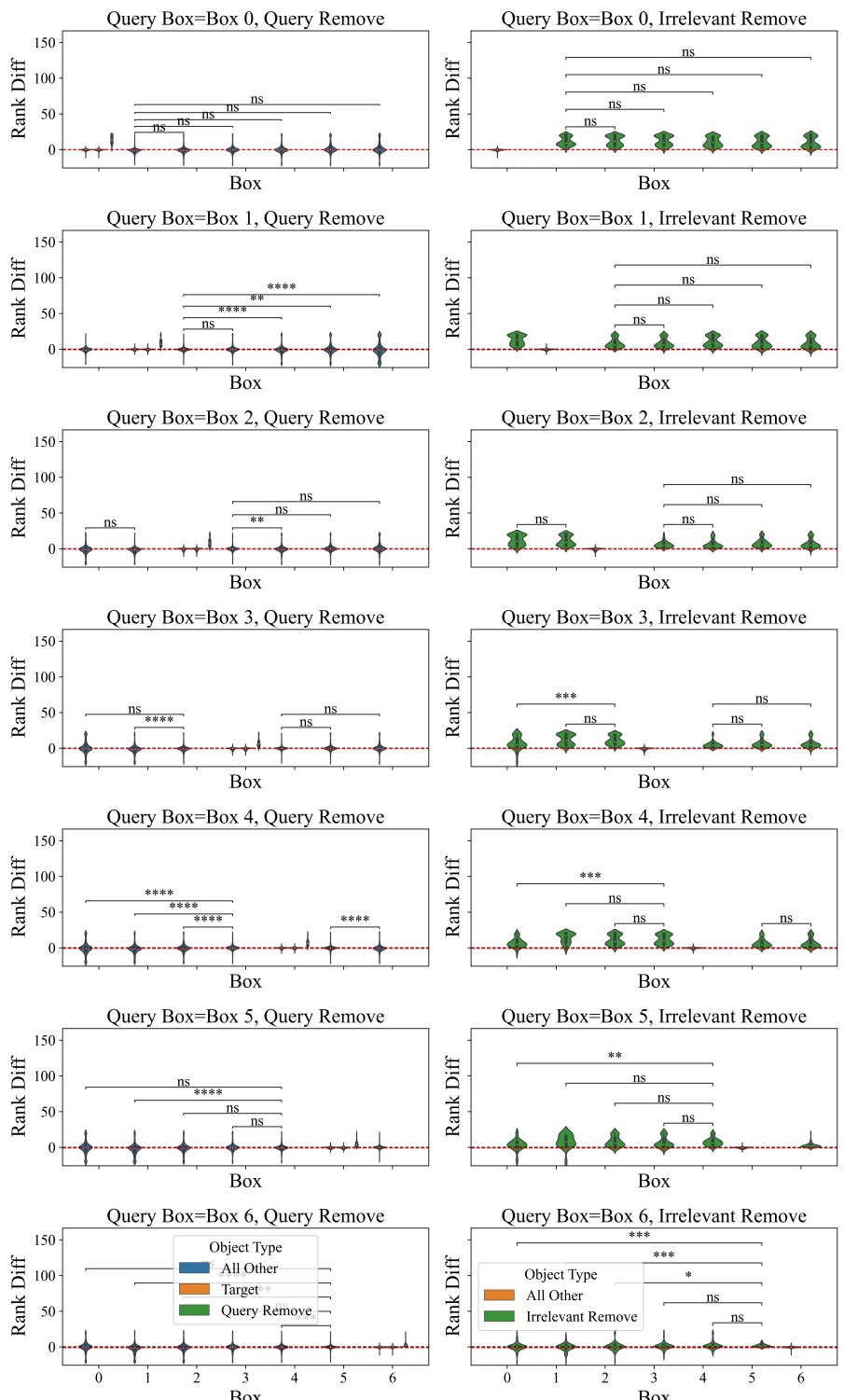

*Figure 23.* REMOVE phrase rank diff split by query box position (CODELLAMA-13B, 2-shot) shows that the effect of the diffused rank increase from **irrelevant REMOVE** is not present in 2-shot setting (right) but still somewhat present for **other** objects, suggesting few-shot learning could alter actual mechanisms. **Black** dotted-line is 0-baseline (i.e. no rank change), and **red** dotted-line is average rank diff across all **other** objects. We use 1-tail Mann-Whitney test for statistical significance. **ns**: $0.05 < p \leq 1$, *: $0.01 < p \leq 0.05$, **: $0.001 < p \leq 0.01$, ***: $0.0001 < p \leq 0.001$, and ****: $p \leq 0.0001$

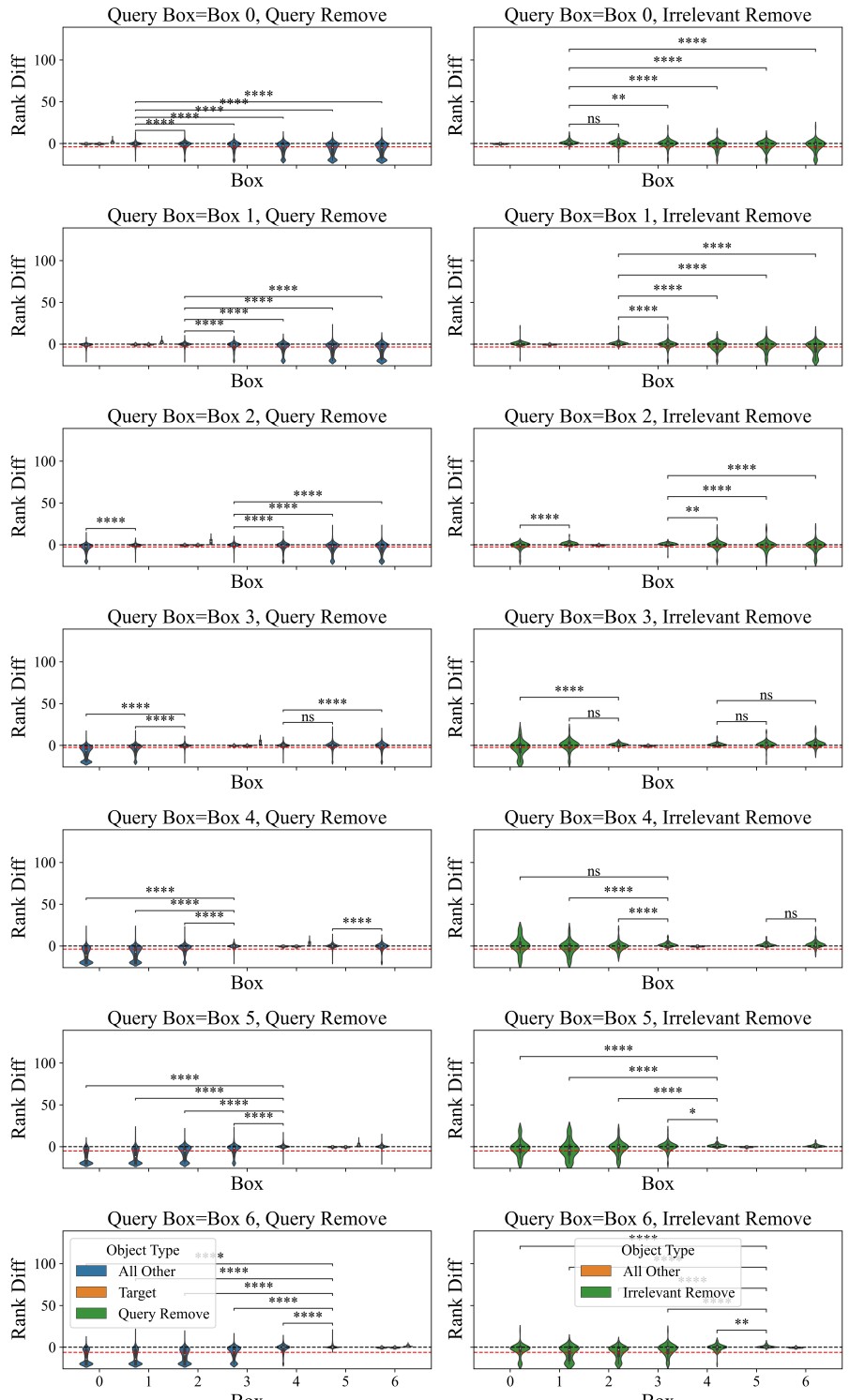

*Figure 24.* REMOVE phrase rank diff split by query box position (GEMMA-2-9B, 2-shot) shows that the effect of the diffused rank increase from **irrelevant REMOVE** is still present in 2-shot setting, different from that of CODELLAMA-13B. This suggests the effect of few-shot learning on task mechanism could be different among models. **Black** dotted-line is 0-baseline (i.e. no rank change), and **red** dotted-line is average rank diff across all **other** objects. We use 1-tail Mann-Whitney test for statistical significance. **ns**: $0.05 < p \leq 1$, *: $0.01 < p \leq 0.05$, **: $0.001 < p \leq 0.01$, ***: $0.0001 < p \leq 0.001$, and ****: $p \leq 0.0001$

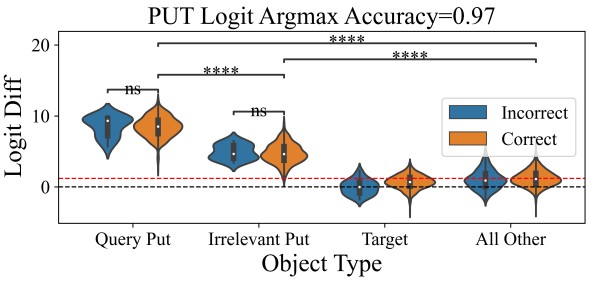
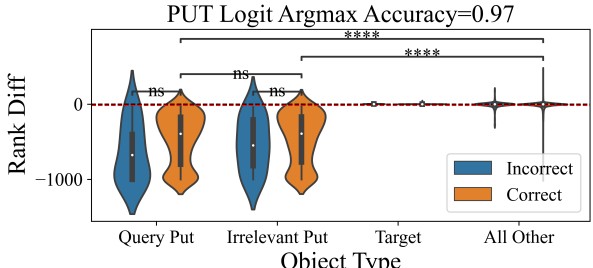

*Figure 25.* Logit (left) and rank (right) difference of object with 1-`PUT` operation that is applied on either the query box or an irrelevant box for CODELLAMA-13B zero-shot. There is a smaller increase in logit for irrelevant `PUT` than query `PUT`. Rank diffs are clipped between $[-1000, 1000]$. We use 2-tail Mann-Whitney test. **ns**: $0.05 < p \leq 1$, *: $0.01 < p \leq 0.05$, **: $0.001 < p \leq 0.01$, ***: $0.0001 < p \leq 0.001$, and ****: $p \leq 0.0001$

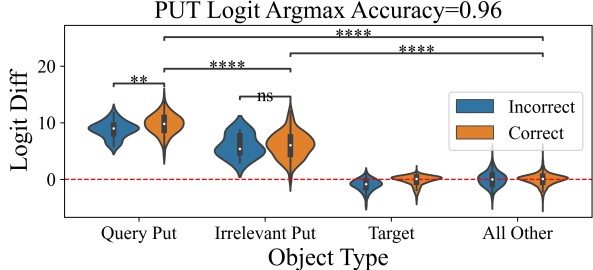
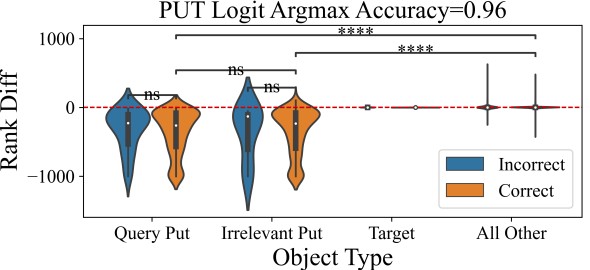

*Figure 26.* Logit (left) and rank (right) difference of object with 1-`PUT` operation that is applied on either the query box or an irrelevant box for CODELLAMA-13B two-shot. Rank diffs are clipped between $[-1000, 1000]$. We use 2-tail Mann-Whitney test. **ns**: $0.05 < p \leq 1$, *: $0.01 < p \leq 0.05$, **: $0.001 < p \leq 0.01$, ***: $0.0001 < p \leq 0.001$, and ****: $p \leq 0.0001$

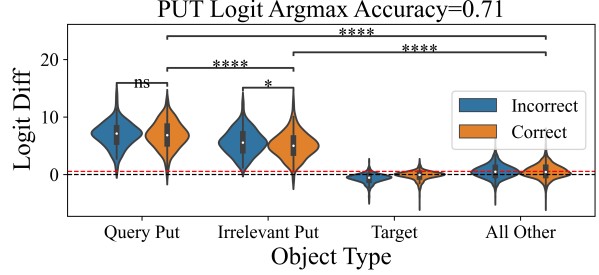
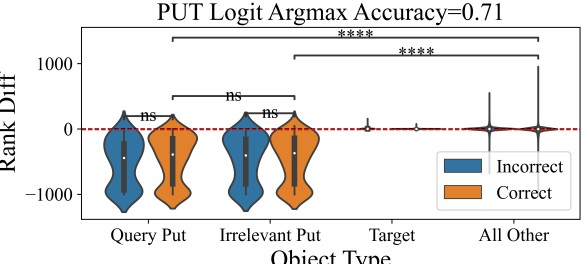

*Figure 27.* Logit (left) and rank (right) difference of object with 1-`PUT` operation that is applied on either the query box or an irrelevant box for GEMMA-2-2B zero-shot. Rank diffs are clipped between $[-1000, 1000]$. We use 2-tail Mann-Whitney test. **ns**: $0.05 < p \leq 1$, *: $0.01 < p \leq 0.05$, **: $0.001 < p \leq 0.01$, ***: $0.0001 < p \leq 0.001$, and ****: $p \leq 0.0001$

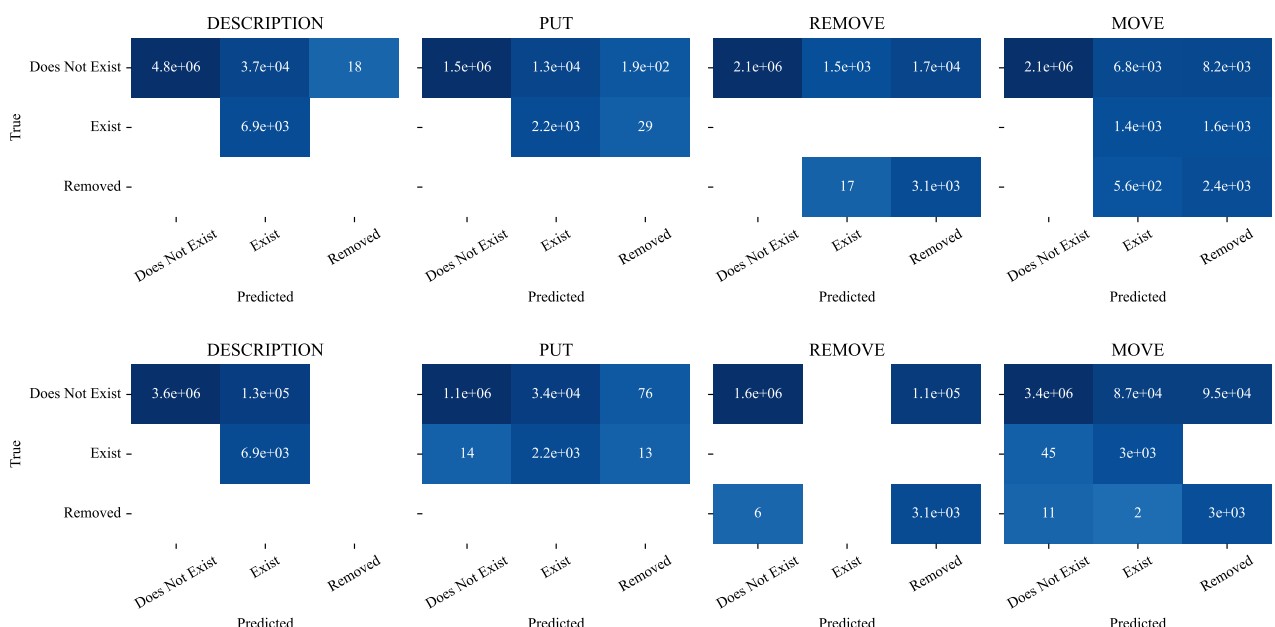

*Figure 28.* Confusion matrix for CODELLAMA-13B (layer 8) ternary probes conditioned on the object token (top) and the box ID token (bottom).

In Fig. 29, we see `Exists/Removed` are close to antipodal right before layer 5, while they are closer to orthogonal to the `Non-exist` class around the same layer (left figure). Around the same layer, we also see `Non-exist→Removed` greater than `Removed→Non-exist` (middle figure). On the right plot around the same layers (0-5), we see the rank of the matrix is ordered by `Removed < Exist < Non-exist`. This confirms our intuition about the structural properties of the probes, that `removed` probe weights are well-approximated by a low-dimensional subspace already present in the `non-exist` weights.

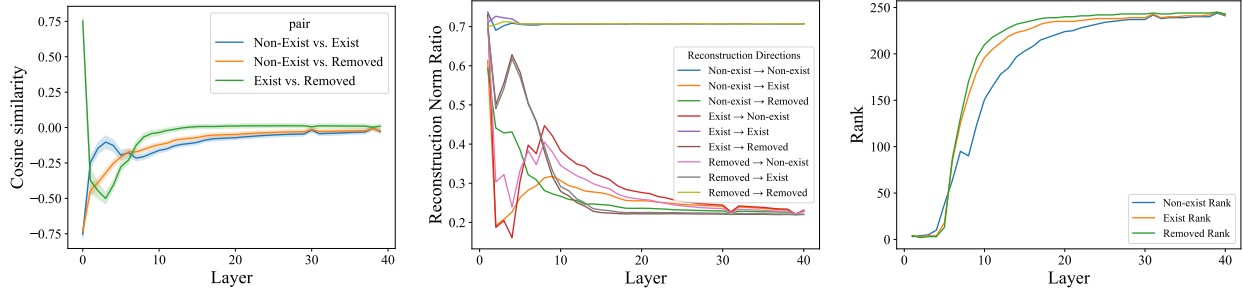

*Figure 29.* Structural analysis of Box ID probe (CODELLAMA-13B) reveals more antipodal representation between `exist/removed` and **removed** signal is in a subspace of **exist**.

### H.9. Ternary Probe Results and Analysis for LLAMA-3.1-70B

In Fig. 30, we show the ternary probe results for LLAMA-3.1-70B conditioned on the object and box ID token. Similar to CODELLAMA-13B, we see high accuracy for **Local Box-Object** in early layers of the model. The maximum accuracy for object probe is around 0.84. This is the highest accuracy obtainable for object probes because MOVE operations make up $\frac{1}{3}$ of all operations as they are uniformly sampled, and randomly predicting over class labels `remove/exist` will yield only 50% accuracy over those datapoints. This accounts for the $\frac{1}{3} * \frac{1}{2} = \frac{1}{6}$ (or around 16%) error rate. The model cannot determine the labels for the MOVE operations at the object tokens because they appear before box IDs, and there are two possible labels for the box-object pairs depending on if the box ID is the first or the second operand within the phrase. For example, in *Move the apple from Box 1 to Box 3*, the label for {(*apple, 1*), (*apple, 3*)} is not decided until the token *1*, which

appears after *apple*.

In Fig. 32 we show probe accuracy across context (same plot as main text Fig. 8), and in Fig. 31, we show structural analysis of the probes (similar to CODELLAMA-13B Fig. 29). In Fig. 33 we show probe confusion matrix across operations for layer 10, where probe performance is the highest. It shows similar pattern as CODELLAMA-13B (Fig. 28). In Fig. 34, we show intervention results with single operations. This result also follow conclusions from the main text Fig. 9.

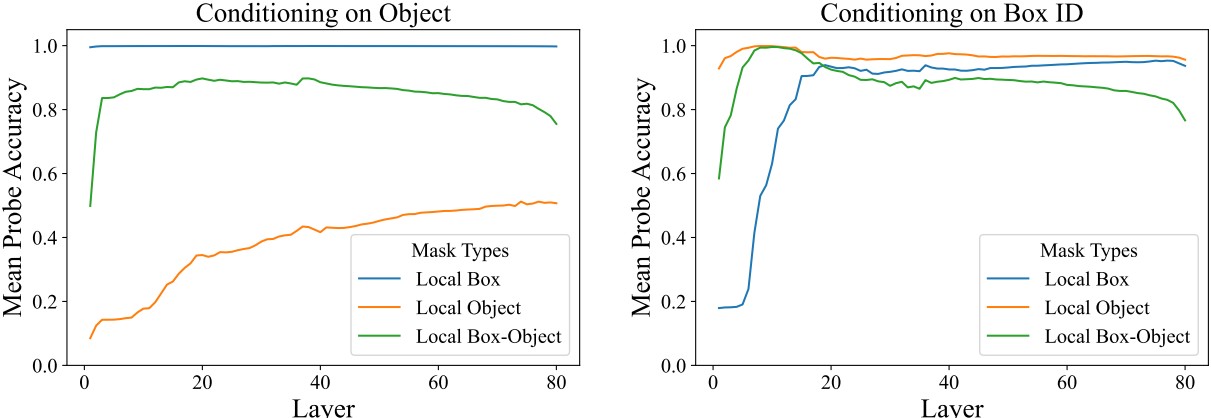

*Figure 30.* Probe accuracy of LLAMA-3.1-70B conditioned on **Object** tokens and **Box ID** tokens suggests both locations contain signal (around layer 5-10) for the "remove-tag" with stronger signal on Box ID.

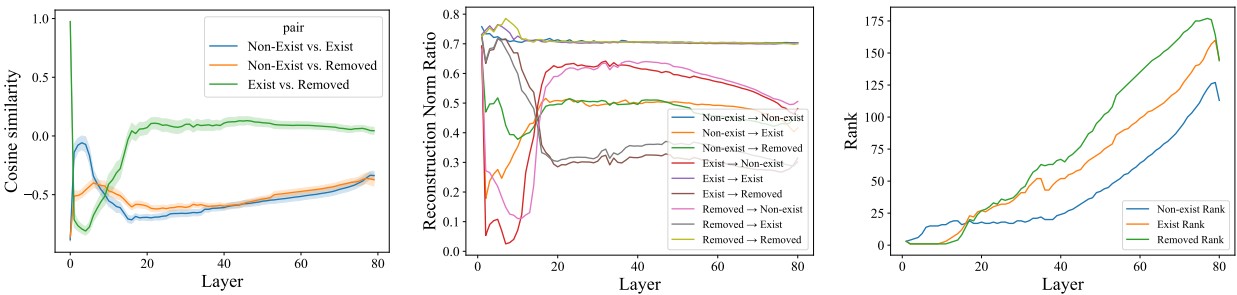

*Figure 31.* Structural analysis of Box ID probe (LLAMA-3.1-70B) reveals closer (antipodal) representation for `exist/removed` probe weights and that **removed** signal is in a subspace of **exist**.

## H.10. 1-**PUT** Intervention Error Analysis

In Sec. 4.2.3 (Fig. 9, PUT, top), nullifying "exist-tag" at the object token in CODELLAMA-13B only results in partial success. When this intervention is applied at the $o_{\text{put}}$, the model stops predicting $o_{\text{desc}}$ as well as $o_{\text{put}}$. The errors fall into the following three types:

- **Enum**: enumerating objects in the order in which they appeared in context. Usually the enumeration starts from the first object in context (i.e. Box 0 description phrase object).

- **OID**: predicting an object one position adjacent to the target object.

- **Other**: predicting another object with no identifiable pattern.

We built an automated script to classify errors into the above three categories. In Fig. 35, we show that across interventions at different single layers at the object token position, **Error (Enum)** occurs about 10% of the time, **Error (OID)** about 80% of the time, and other errors around 10% of the time. This suggests that there could be multiple competing mechanisms that the model uses to predict object tokens, while there is only one correct mechanism for tracking. The high **Error (OID)** rate

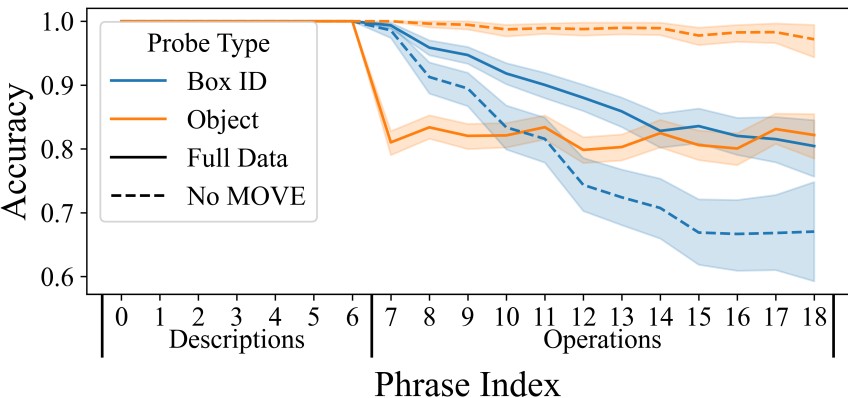

*Figure 32.* Ternary probe accuracy (LLAMA-3.1-70B) conditioned on Object and Box ID across phrase index indicates the "tag" signal weakens across context at Box ID token, and not at Object token.

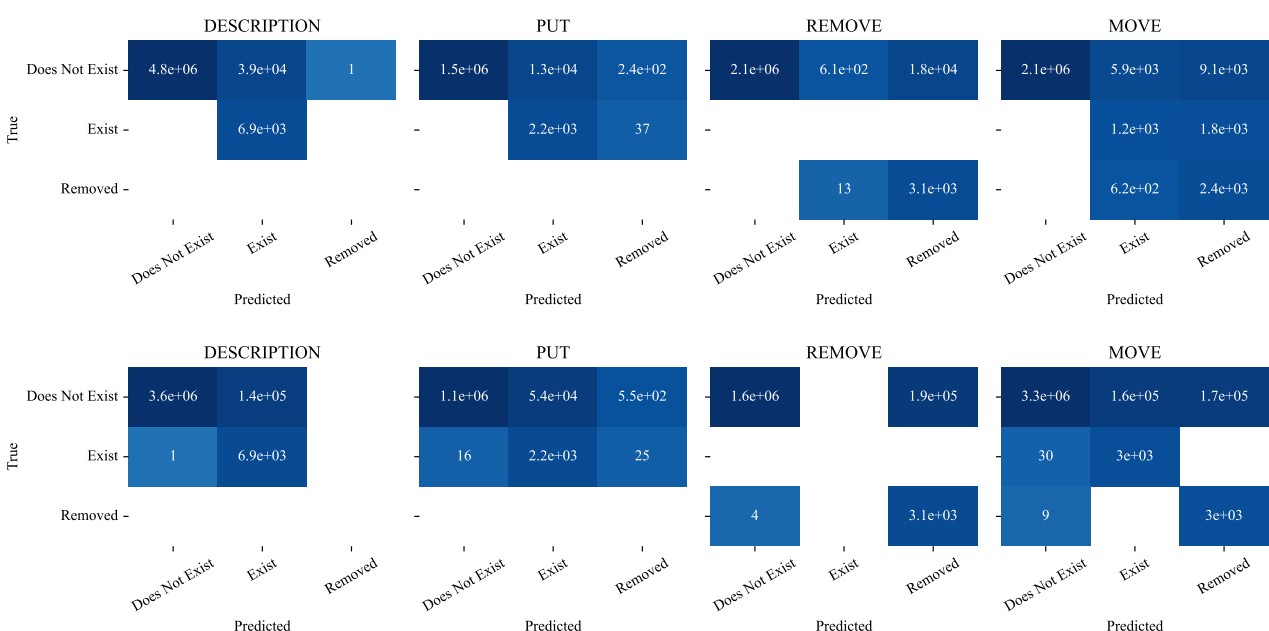

*Figure 33.* Confusion matrix for LLAMA-3.1-70B (layer 10) ternary probes conditioned on the object token (top) and the box ID token (bottom).

across layers indicates that the model up-weights probabilities of all objects adjacent to the target object, and such effect diminishes as objects are located farther away from the target object. This gaussian-like effect on the up-weighting of entity probabilities across their positional IDs is also seen in Gur-Arieh et al. (2025). Most of the time, the model still predicts the same number of objects as before the intervention, suggesting that tracking the number of target objects exists as a separate mechanism independent of the identity of the objects that we capture with our probe.

### H.11. Alternative Box ID Token Interventions with Probes

In Sec. 4.2.3 (Fig. 9, REMOVE/MOVE-OUT, bottom), we show single-layer nullification intervention does not work for REMOVE/MOVE-OUT at the Box ID token. Here we show additional negative results (Fig. 36) for interventions at the Box ID token: null-ing first-n (intervening from first to n-th layer) and last-n (intervening from n-th to last layer) results for the same experiment, as well as negating "remove-tag" (push across the decision boundary) or boosting "exist-tag", following

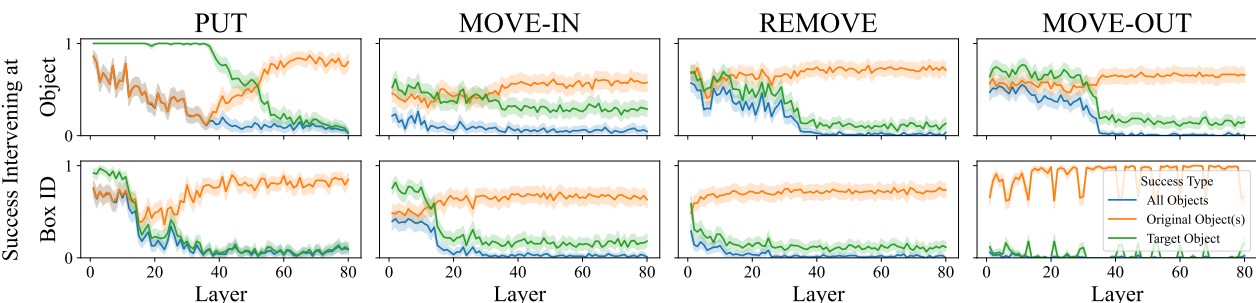

*Figure 34.* Single layer intervention results for nullifying the remove tag (in `REMOVE/MOVE-OUT`) and exist tag (in `PUT/MOVE-in`) in LLAMA-70B suggests that despite high probing accuracy, causally relevant remove tag resides in object tokens, while exist tag is in box ID tokens. Due to computational constraints, we use 100 examples for each operation in this experiment.

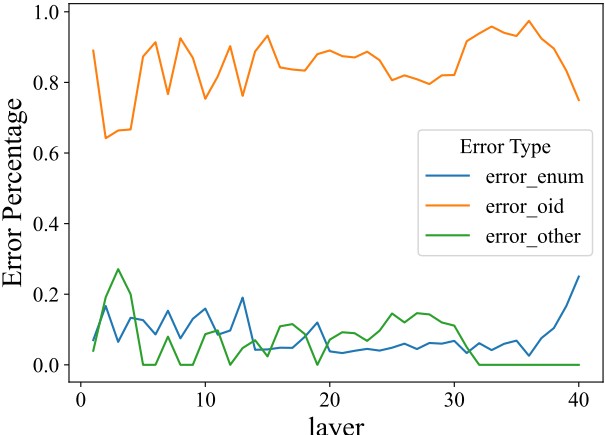

*Figure 35.* Error types across intervening examples with a single `PUT` on object probe suggest that even when the "exist tag" from object tokens are removed, positional information (**Error (OID)**) continues to have big effect on the prediction, causing models to predict objects adjacent to the `PUT` object.

Ravfogel et al. (2021) (Fig. 37). In all of the figures, we see that models are not able to successfully process target objects (solid lines remain at the bottom of the plot). This further supports our hypothesis that causally efficacious signals for "remove-tag" are not present at Box ID.

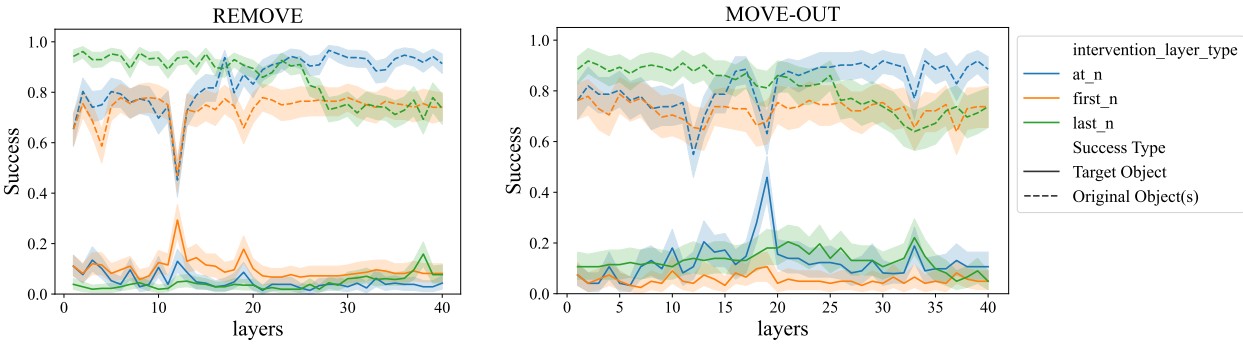

*Figure 36.* Intervention results on CODELLAMA-13B on first or last $n$ layers on "remove tags" at Box ID token. We are not able to successfully recover the model from predicting the removed object, supporting our claim that causally efficacious signals for "remove tags" are **not** at Box ID token.

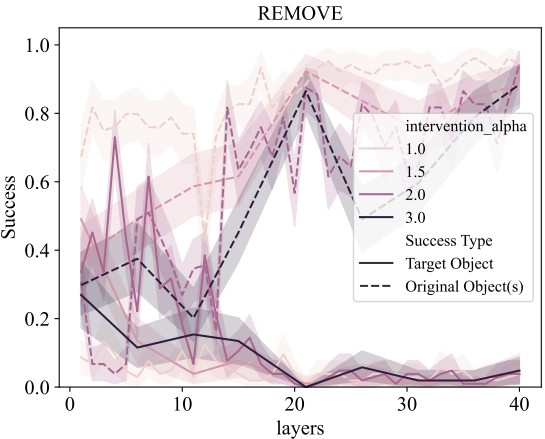
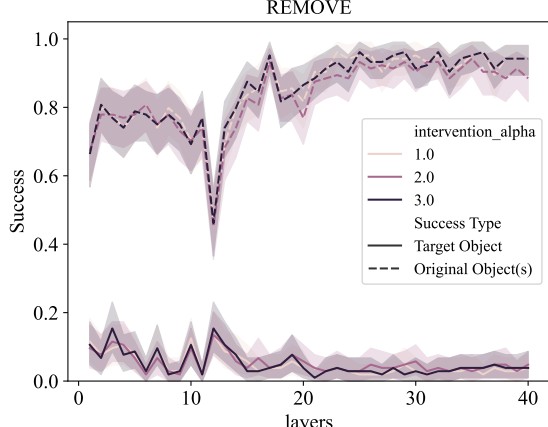

*Figure 37.* Intervention results on CODELLAMA-13B negating the "remove tag" (left) or boosting the "exist tag" (right) at Box ID token. We are not able to successfully recover the model from predicting the removed object.

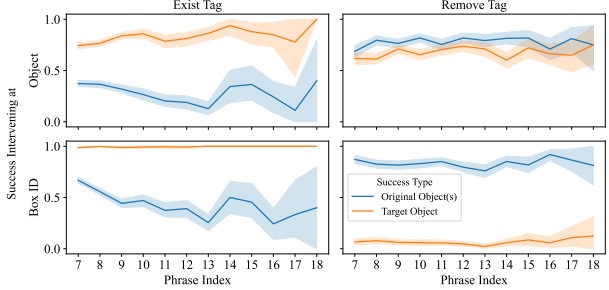
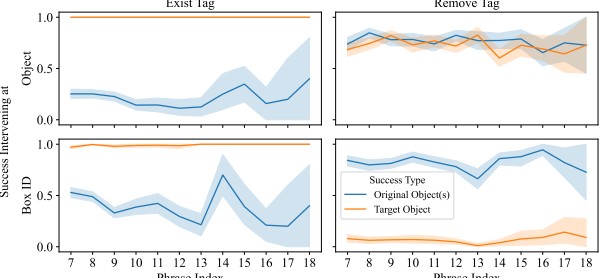

*Figure 38.* Single-layer intervention (at layer three) success rates across phrase index of where the query operation occurs (CODELLAMA-13B). Left plot shows full results and right plot shows results without MOVE operation. Results confirm the ineffectiveness of intervention for remove tag at the box ID token (bottom right at either side). It also shows that the success rate for exist tag is affected by where the query operation appears in the context (blue line fluctuates in first and third column), while not so much for the remove tag (blue and orange lines are flat on top right plot at either side).

## H.12. Intervention Across Context

In Sec. 4.2.2, we showed that probe accuracy decreases linearly at the Box ID token but remains steady at the object token. Here we perform intervention on the tags that are at different relative positions of the example to establish the causal effect of the diminishing tag information on model predictions. We perform intervention on behaviorally successful examples from CODELLAMA-13B in altForm_default (test split). We filter for examples where there is only a single query operation, and check intervention success rate across the phrase index of the query operation within the example. There are seven description phrases and up to twelve operation phrases. The phrase index corresponds to the position where the operation occurs within the example. We perform single-layer intervention at layer three since it has the highest success rate (Fig. 9). In Fig. 38, we show intervention results with all four operations (left) and without MOVE operation (right; since probe signal at the object token is undetermined, see Sec. 4.2.2). Bottom right plots of either side confirms that intervention at the box ID token is ineffective for remove tag. For exist tag, there appears to be a U-shaped accuracy curve across context, similar to findings in Gur-Arieh et al. (2025), whereas remove tag intervention at the object token remain relatively constant throughout context (top right on either side). Removing the MOVE operation also allows us to see that intervention for the exist tag always results in desirable results with the target object, but the failure is often in maintaining the original object(s) intact.

## H.13. Cumulative Remove Tags

In Sec. 3, we showed evidence for the model tracking local states, rather than cumulatively tracking states. Within the ternary probe setting, we additionally test whether remove tags are cumulative across phrases (i.e. in ... *Remove the apple*

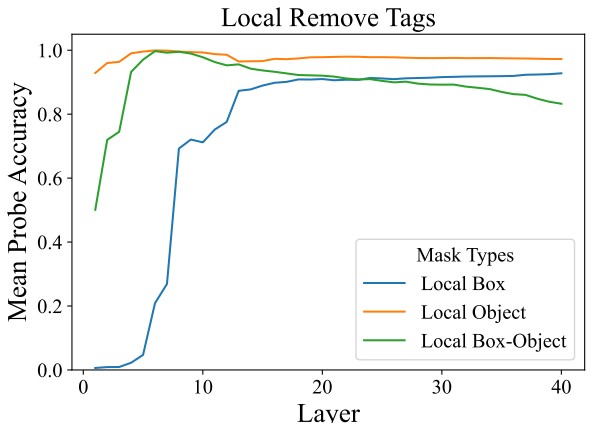 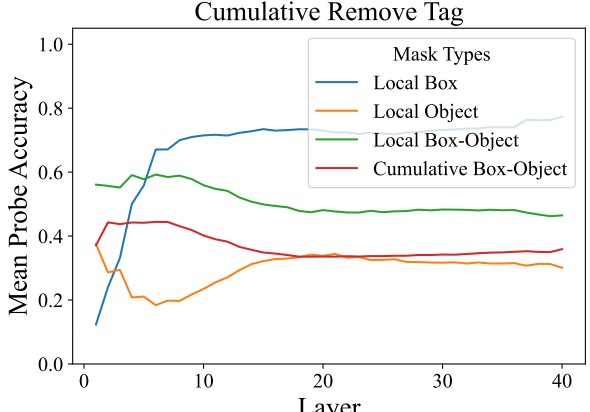

*Figure 39.* Probe accuracy of CODELLAMA-13B completion conditioned on Box ID tokens with local states (left) and cumulative states (right). Crucially, **cumulative box-object** line and **local box-object** line on the right are much lower than **local box-object** line on the left, suggesting models do not accumulate "remove-tags" across phrase boundaries.

*from Box 1. Remove the banana from Box 2. ...*, "remove-tag" for (*apple, Box 1*) would be detectable from Box 2 if remove tags were cumulative).

We train the probes the exact same way as the local ternary probes (Details in Appendix H.6) and compute metrics by averaging over probes that are relevant. Specifically, we keep **local box**, **local object**, **local box-object** masks the same for both label types for fair comparison (see Sec. 4.2.2 for details), and additionally compute **cumulative box-object** for cumulative probes. With cumulative box-object mask, we average over all box-object pairs that has been mentioned in the context (e.g. {(*apple*,0), (*peach*,1), (*clock*,2), (*jar*,2), (*television*,3), (*brain*,4), (*book*,5), (*pin*,6), (*watch*,1)} for the REMOVE phrase in Example 2.1). Even though the cumulative box-object mask aggregates over more probes than local box-object, it measures the probe accuracy over all box-object pairs the cumulative probes should learn. If models were to accumulate "remove-tag" across the context, this metric would be high.

We show the results in Fig. 39. We see model performs nearly at random for cumulative box-object metric(random uniform prediction accuracy = 0.33). All other metrics drop as well (comparing the same lines in left and right plots), indicating the noise in the signal corrupts learning the box and objects in the current phrase. This experiment confirms that "remove-tags" are *not* cumulative across phrase boundaries.

### H.14. DCM for 1-REMOVE

In Sec. 4.2.1 and Appendix H.3 we show that there is likely a box ID specific mechanism that suppresses the logit of the removed object. Such effect also resembles that of positional signals in entity binding (Appendix H.4). We also see that intervening on the object token results in expected prediction changes, but not on the Box ID token (Sec. 4.2.3). In this experiment, we further explore the information this mechanism might use for communicating with downstream components. We constructed DCM counterfactual sentences to see if the "remove-tag" is picked up (anywhere in the REMOVE phrase) by query tokens as positional information similar to that of the DESCRIPTION/PUT circuit. Here is an example sentence and its counterfactual (Example H.14):

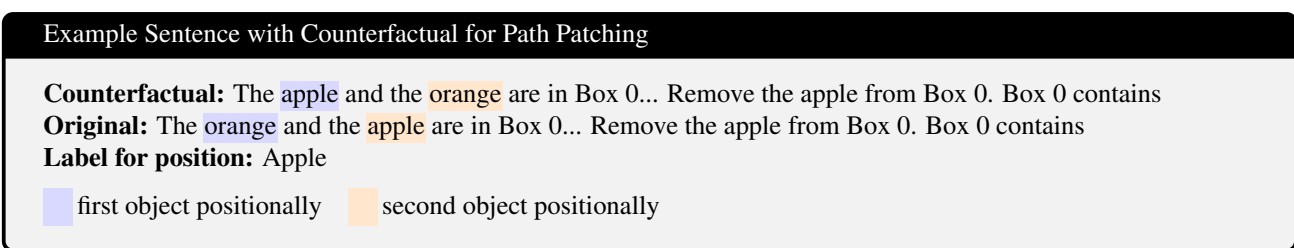

If REMOVE were to operate on positional information, patching the "remove-tag" from counterfactual to original sentence

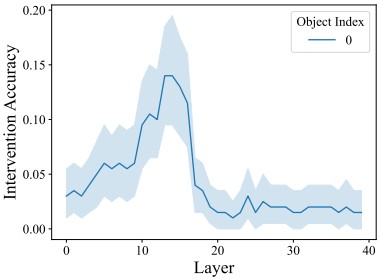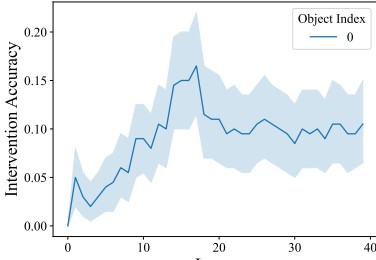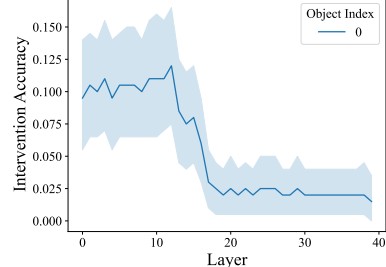

*Figure 40.* DCM results patching part of the REMOVE phrase (CODELLAMA-13B) suggests that positional information is not the main signal used in communication with downstream components. Left to right columns are patching at a single layer, first-$n$ layers, and last-$n$ layers.

would mean that the model is placing a "remove-tag" at the first object (which is *orange* in the original sentence), which would result in model predicting *apple*. When patching, we tried patching the following token(s)

1. the object token (second *apple*) in the REMOVE phrase.

2. the Box ID token (second *apple*) in the REMOVE phrase.

3. the period token at the end of the REMOVE phrase.

4. the entire REMOVE phrase.

We show the results across 200 examples for CODELLAMA-13B for intervening on the entire REMOVE phrase in Fig. 40 as it showed the best intervention accuracy, which is still only around 15%. The peak of accuracy around layer 13 in the left plot (single-layer intervention) is similar to what we observe in Fig. 4b,c, which suggests that positional information is present. However, the best intervention accuracy of 15% suggests that this information is not the main signal used in communication with downstream components.

# I. Prompting Structure

### I.1. Completion

In the completion setting, we simply provide the state description and have model complete the sentence after pre-filled query phrase *Box X contains*. This is analogous to the original setup from Kim & Schuster (2023). Unless otherwise mentioned, "0-shot" in the paper is "0-shot" completion setting. For most of the experiments, we filter out examples with an empty query box, so we can additionally pre-fill the query phrase with *the*. See an instance of the completion prompt in Example D.1.

### I.2. 0-shot with instruction

> **Prompt for 0-shot evaluation with instruction**
>
> Given the description after "Description:", write a true statement about a box and its contents according to the description after "Statement:". If a box is empty, write "Box X contains nothing".
>
> Description: {CONTEXT}
> Statement: Box {QUERY_BOX} contains

## I.3. 2-shot with instruction

> **Prompt for 2-shot evaluation with instruction**
>
> Given the description after "Description:", write a true statement about a box and its contents according to the description after "Statement:". If a box is empty, write "Box X contains nothing".
>
> Description: The car is in Box 0, the cross is in Box 1, the bag and the machine are in Box 2, the paper and the string are in Box 3, the bill is in Box 4, the apple and the cash and the glass are in Box 5, the bottle and the map are in Box 6.
> Statement: Box 3 contains the paper and the string.
>
> Description: The car is in Box 0, the cross is in Box 1, the bag and the machine are in Box 2, the paper and the string are in Box 3, the bill is in Box 4, the apple and the cash and the glass are in Box 5, the bottle and the map are in Box 6. Remove the car from Box 0. Remove the paper and the string from Box 3. Put the plane into Box 0. Move the map in Box 6 to Box 2. Remove the bill from Box 4. Put the coat into Box 3.
> Statement: Box 2 contains the bag and the machine and the map.
>
> Description: {CONTEXT}
> Statement: Box {QUERY_BOX} contains

## I.4. 2-shot with instruction enumerating all box contents

> **Prompt for 2-shot evaluation with instruction**
>
> Given the description after "Description:", write a true statement about all boxes and their contents according to the description after "Statement:". If a box is empty, write "Box X contains nothing".
>
> Description: The car is in Box 0, the cross is in Box 1, the bag and the machine are in Box 2, the paper and the string are in Box 3, the bill is in Box 4, the apple and the cash and the glass are in Box 5, the bottle and the map are in Box 6.
> Statement: Box 0 contains the car, Box 1 contains the cross, Box 2 contains the bag and the machine, Box 3 contains the paper and the string, Box 4 contains the bill, Box 5 contains the apple and the cash and the glass, Box 6 contains the bottle and the map.
>
> Description: The car is in Box 0, the cross is in Box 1, the bag and the machine are in Box 2, the paper and the string are in Box 3, the bill is in Box 4, the apple and the cash and the glass are in Box 5, the bottle and the map are in Box 6. Remove the car from Box 0. Remove the paper and the string from Box 3. Put the plane into Box 0. Move the map in Box 6 to Box 2. Remove the bill from Box 4. Put the coat into Box 3.
> Statement: Box 0 contains the plane, Box 1 contains the cross, Box 2 contains the bag and the machine and the map, Box 3 contains the coat, Box 4 contains nothing, Box 5 contains the apple and the cash and the glass, Box 6 contains the bottle.
>
> Description: {CONTEXT}
> Statement: Box {QUERY_BOX} contains

