# OpenReview forum: "Do Language Models Track Entities Across State Changes?"
_ICML.cc/2026/Conference — ICML 2026 regular_

### Official Review · Reviewer_Me3u · 2026-03-05

**Soundness:** 3
**Presentation:** 3
**Significance:** 3
**Originality:** 3
**Overall Recommendation:** 5
**Confidence:** 3

**Summary:**

This paper investigates how transformer language models perform entity tracking (ET) when the task involves multiple state-changing operations — PUT, REMOVE, and MOVE — expressed in natural language. The central finding is that LMs do not build world states incrementally as they process context; instead, they dynamically consolidate relevant information only when the query becomes apparent. The authors further characterize the circuits underlying individual operations, finding that PUT resembles known entity binding circuits while REMOVE relies on a fragile "global removal tag" that predicts several behavioral failure modes.

**Compliance With Llm Reviewing Policy:**

Affirmed.

**Key Questions For Authors:**

1. Most of the deep mechanistic work is performed on CodeLlama-13B. While you show behavioral consistency in Llama-3-70B/405B, did you find evidence of the same "global suppression tag" in non-code-specialized models like the original Llama-2 or Mistral?

2. Your intervention (nullifying the removal tag) recovered 42% of failure cases. For the remaining 58% of cases where the model still failed despite the intervention, what do you hypothesize is the secondary failure mechanism?

**Limitations:**

* All target objects are single tokens, which may not reflect real-world entity tracking complexity
* The global remove hypothesis is specific to datasets where each object appears in at most one box; more complex scenarios may require different mechanisms
* Computational constraints meant some experiments (e.g., LLaMA-70B interventions) used only 50 examples, which is a small sample

**Strengths And Weaknesses:**

## Strengths

* Clear, testable hypotheses (H1-H4) with well-designed probe experiments to distinguish between them.
* Findings replicate consistently across multiple model families and scales.
* Global remove hypothesis is compelling. Predicting novel failure modes from mechanistic analysis and then confirming them behaviorally is a strong validation loop.
* Practical implications are well-articulated: CoT explanation, finetuning directions, and a proof-of-concept intervention.

## Weaknesses
* REMOVE mechanism remains only partially characterized; circuit tracing yielded negative results, leaving a meaningful gap.
* The intervention fix for "Re-introducing Removed Object" recovers only ~42% of failure cases with limited explanation of why.
* Findings are demonstrated primarily on a synthetic box dataset — generalization to less structured natural language is not established

---

> ### Author Rebuttal · Authors · 2026-03-31
>
> Thank you for your positive review and feedback! We include figures [here](https://anonymous.4open.science/r/et-across-state-changes-rebuttal-BF06/) and references them in-line.
>
> # W1
> We agree that the REMOVE mechanism is only partially characterized (as noted in the discussion), and there is room for future work. However, we believe that our work makes meaningful steps especially when this is a problem that the current interpretability methods cannot be straightforwardly applied to.
>
> First, methods based on counterfactual inputs are difficult to apply due to the difficulty in designing counterfactual inputs that isolate removal operations. This is because most counterfactuals lead to illicit operation sequences—we would appreciate suggestions from the reviewer in this direction if any. Second, activation patching methods rank heads (mostly) by the logit of the target object, but what we need to look at in the removal scenario is the _decrease_ of logit. Patching results based on logit decrease were hard to interpret, which likely stems from the fact that the suppression mechanism works by relative decrease of logit compared to other objects, not absolute decrease. This becomes evident when comparing logits of other objects to the removed objects: in **main_fig.png**, we plot the logit and rank differences (CodeLlama-13B) of the queried & irrelevant removed objects, target object (object in the query box not removed that should be predicted), and other objects in the context (n=500). We see that the removed objects’ logits (query remove) on average does not change, but relative to positive changes in all other objects, this non-change effectively works as suppression (as also shown by the increase in their rank).
>
> The black dotted line is 0-baseline (no diff), and the red dotted line is avg. rank diff across all `other` objects in the correct cases. We will add this analysis to the next revision of the manuscript.
>
> # W2&Q2
> This is a great point! We believe intervention was limited for the “reintroducing removed object” experiment because there is also an additional order issue at play where consolidation of multiple operations needs to happen in sequential order for the answer to be correct. The other two scenarios do not have this ordering consideration, so we conducted intervention for both of them (where the prediction is that intervention will recover the performance better). The same intervention (object probe, single layer intervention at L1) results in 72.8% recovery rate for no-op removal (n=256) and 65.6% (n=183) for shared-label objects in multiple boxes. We believe these additional intervention results do suggest that the global remove mechanism is the culprit for the majority of the failures in these cases; we will add these results and discussion to the next revision of the manuscript.
>
> # W3
> Please refer to our response to Reviewer J4Ka (first bullet point).
>
> # Q1
> Thanks for this suggestion! We applied the new logit/rank diff experiment discussed above (that shows suppression via relative logit decrease) to 3 additional, non-code-specialized models: Llama-2-7b/13b and Mistral-7b (**see Figures in link**). We find that only Llama-2-7b shows an insignificant difference between `irrelevant remove` vs. `other`, suggesting that the global removal hypothesis often holds for non-code models as well.
>
> # L1
> Our analyses may be generalizable to multi-token objects using the first token of the object as a proxy (since the completion of the object label after predicting the first token would be straightforward), but we agree that this leaves room for interesting future work. One version of the dataset from Kim and Schuster (2023) does have multi-token objects with different modifiers (“red apple” vs. “green apple”), which we could look into.
>
> # L2
> The global remove hypothesis is not specific to datasets that have objects with the same label appear in at most one box, although a global removal mechanism, despite being degenerate, will allow a model to perform behaviorally well on such a dataset. To clarify, our claim is that this property of the dataset actually MASKS the failure modes that stem from global remove, because the outcome of global removal is indistinguishable from the outcome of local removal, and therefore the models can achieve high behavioral accuracy with the (degenerate) global removal mechanism. The new scenarios in Section 4.4 aims to exactly illustrate this by introducing new test cases NOT in the original dataset that a model with global removal mechanism would behaviorally fail on. Thus, your comment “more complex scenarios may require different mechanisms” is exactly the point we try to illustrate with introducing these additional scenarios, and to highlight the limitation of the mechanism we find in the models that is masked by the limitations of the dataset.
>
> # L3:
> We find similar qualitative trends with n=100 (**100example-2shot-70b.png**) for Llama-70B (0-shot).

---

> > ### Author Rebuttal · Reviewer_Me3u · 2026-04-06
> >
> > My concerns have been adequately addressed.

---

### Official Review · Reviewer_H6mW · 2026-03-11

**Soundness:** 3
**Presentation:** 2
**Significance:** 3
**Originality:** 3
**Overall Recommendation:** 4
**Confidence:** 3

**Summary:**

This paper investigates how LLMs perform entity tracking when entities undergo state-changing operations such as PUT, REMOVE, and MOVE. The authors analyze whether LMs maintain and update an internal world state across tokens and layers when processing such operations. Through mechanistic analyses and probing experiments, it is interesting that they find that LLMs do not build world states incrementally across tokens or layers, but simply retrieve and aggregate relevant information at the last token when the query
becomes evident (clear). But they find LLMs tend to do the ET through the retrieval of target tokens in parallel across the context.

**Compliance With Llm Reviewing Policy:**

Affirmed.

**Final Justification:**

Good Paper, accepted.

**Key Questions For Authors:**

No

**Limitations:**

Yes

**Strengths And Weaknesses:**

Strengths: The paper provides a clear mechanistic analysis of how LLMs implement entity tracking under state changing operations. By applying circuit tracing and patching, they identify distinct groups of attention heads responsible for different roles and systematically compare the circuits for different operations. Then they do the intervention experiment to prove the circuit they find is right.

Weaknesses: The paper mainly analyzes PUT and DESCRIPTION. Since the experiments are fairly limited in terms of operations and settings, it is hard to tell whether the same mechanisms would still hold for more complex state changes or other models.

---

> ### Author Rebuttal · Authors · 2026-03-31
>
> Thank you for your positive review and feedback!
>
> Weaknesses: Regarding generalization to other models, our experiments include CodeLlama 13B, Llama 70B, Llama 405B, Gemma 2B, and even more models in the analysis of REMOVE (Qwen family, Gemma family, Llama Family in Appendix G2). The operations we analyze may be few in number but we believe they form the basis of complex state changes since they can be composed/stacked to describe problems of arbitrary complexity. Furthermore, while additional mechanisms may be involved in multiple query operations (e.g. operation order resolution), one of our main findings (models do not build world state across context or local states across layers) is a generalizable takeaway that is applicable to problems with more complex state changes. If you have further reservations about the limitations of the evaluation dataset itself, please refer to our response to Weaknesses pointed out by Reviewer J4Ka.

---

> > ### Author Rebuttal · Reviewer_H6mW · 2026-04-03
> >
> > Good Rebuttal.

---

### Official Review · Reviewer_J4Ka · 2026-03-13

**Soundness:** 3
**Presentation:** 3
**Significance:** 2
**Originality:** 2
**Overall Recommendation:** 5
**Confidence:** 2

**Summary:**

The paper investigates the mechanisms of Entity Tracking (ET) in Large Language Models (LLMs) when processing state-changing operations expressed in natural language. Using mechanistic interpretability techniques like linear probing, path patching, and circuit analysis, the authors examine how models like CodeLlama, Llama 3, and Gemma handle operations such as PUT, REMOVE, and MOVE.

**Compliance With Llm Reviewing Policy:**

Affirmed.

**Key Questions For Authors:**

- The paper posits that models aggregate information only once the query becomes evident. If the query were less specific (e.g., "Describe the current state of all boxes"), would the model still accurately extract states, or does the mechanism strictly depend on a localized query trigger?
- Probing experiments show Box ID detection accuracy declines as operations increase. In much longer contexts (e.g., 50 operations), would this retrieval-based mechanism fail completely due to the dilution of attention scores?
- Given the "brittle" global nature of the removal tag, would the model be unable to process complex conditional logic—such as "Remove all fruits, but keep the red apple"—since the suppression might apply to the "apple" token globally?
- If there were multiple identical entities (e.g., three "apples" in different boxes), would the token-based global removal mechanism erroneously "erase" all remaining apples when instructed to remove only one?
- The paper suggests CoT improves performance by shortening the context for tag signals. Could CoT also be interpreted as forcing the model to perform "local aggregations" at the end of each generated sentence, thereby externally simulating an incremental tracking process?

**Limitations:**

see weakness and questions

**Strengths And Weaknesses:**

## Strengths
- The paper goes beyond behavioral observation to identify specific attention head groups (Groups A/B/C/D) and subspaces responsible for entity tracking.
- The authors use their mechanistic findings to successfully predict three new behavioral failure modes, demonstrating a tight loop between internal analysis and external behavior.
- This paper tackles the foundational question of whether Transformers are capable of incremental world modeling or are limited to parallel retrieval heuristics, providing strong evidence for the latter.

## Weaknesses
- The study is confined to a specific "box-and-object" dataset. While this provides control, it remains to be seen if these mechanisms generalize to more complex, abstract, or non-physical state changes
- The research relies on a highly structured synthetic dataset (the "box dataset") with fixed natural language templates, such as "The [object] is in Box [ID]".

---

> ### Author Rebuttal · Authors · 2026-03-31
>
> Thank you for your positive review and feedback!
>
> - **Weaknesses**:  Although the dataset is synthetically generated, we would like to highlight that the dataset we adopted is thus far one of the most naturalistic entity tracking dataset that simultaneously allow us sufficient control for mechanistic analyses. Compare, for instance, our dataset with prior work on entity binding or tracking such as the one adopted in Li et al (2025) that does not use natural language at all, or in Wu et. al., (2025) that uses programming languages. Yet, these studies still contribute significant insights about how language models can perform tracking. Our work continues this effort, and we believe the dataset we adopted in fact helps bring mechanistic work a step closer to tracking problems expressed in natural language. Of course, if the reviewer has other datasets we missed that are more naturalistic and allow for controlled interpretability studies, we would appreciate pointers and would be excited to expand our future studies.
> - **Q1**: This is a great question! Kim and Schuster (2023)’s original evaluation setup in fact required models to describe the content for all boxes. We found that this increases behavioral accuracy for larger models but decreases accuracy for smaller models. Regarding mechanisms, note that at every part of the phrase where the model needs to predict the contents of the box, a localized query trigger is present (i.e. “Describe the content of all boxes: **Box 0 contains** __, … **Box 4 contains** ___…”), so we would expect our mechanistic analyses to still generalize.
> - **Q2**: Yes, we expect Box ID-based retrieval mechanisms to fail in a longer context. In Appendix G6, we also show similar degradation of probe accuracy across context for Llama-70B. While we have not conducted probing in longer contexts, Kim and Schuster (2023)’s behavioral results show substantial degradation of performance over increasing number of operations, which furthermore suggests that this would be the case.
> - **Q3**: We believe this is likely to be true. Based on our analyses of failure models presented in Section 4.4. Although the particular example you provided requires a few other capacities (specifically, comprehension of the universal quantifier “all” and the taxonomic relationship between the word fruit and its possible hyponyms including apple), but failures in the “Shared-label Objects in Multiple Boxes” and “Re-introducing Removed Object” scenarios suggest that the models will also fail on your example. In the shared-label objects scenario we see that the global suppression of “apple” leads to erroneous removal of apples from all other boxes, and in the re-introduction scenario, we see that resolution across multiple operations is difficult. Since “Remove all fruits, but keep the red apple” requires these capacities, we would predict failure. You may also be interested in Kim and Schuster (2023)’s “Ambiguous Reference” tracking dataset where they reported behavioral failure on tracking scenarios like “Box 1 contains the green apple. Box 2 contains the red apple …. Remove the apple from Box 2.”, where the model has to resolve the ambiguous reference by the word “apple” contextually.
> - **Q4**: Yes, this is exactly one of our predicted failure scenarios (“shared-label objects in multiple boxes”) in Section 4.4, and behaviorally we do see this happening in CodeLlama 13B and Llama 70B.
> - **Q5**: Yes, that is our interpretation.
>
> Citations:
> - Kim, N. and Schuster, S. Entity tracking in language mod-
> els. In The 61st Annual Meeting Of The Association For
> Computational Linguistics, 2023.
> - Li, B. Z., Guo, Z. C., and Andreas, J. (how) do language
> models track state? In Forty-second International Con-
> ference on Machine Learning, 2025.
> - Wu, Y., Geiger, A., and Milli` ere, R. How do transformers
> learn variable binding in symbolic programs? In Forty-
> second International Conference on Machine Learning, 2025.

---

> > ### Author Rebuttal · Reviewer_J4Ka · 2026-04-03
> >
> > I remain my positive score

---

### Official Review · Reviewer_BhY1 · 2026-03-13

**Soundness:** 3
**Presentation:** 3
**Significance:** 3
**Originality:** 2
**Overall Recommendation:** 4
**Confidence:** 3

**Summary:**

The paper argues that LLMs do not maintain an incrementally updated global world state for this entity-tracking task. Instead, the author interprets their probing results as showing that models mainly retrieve and compose query-relevant information when the final query is processed: global probes for full object locations are weak, while local probes for whether an object is in the queried box are much stronger. Mechanistically, the paper analyzes individual operations. For PUT, path patching suggests a circuit broadly similar to prior entity-binding circuits, though mostly with different heads. For REMOVE, the author has found that the model uses a brittle object-level “remove tag” that suppresses the removed object at query time, rather than updating a clean box-level state; MOVE is interpreted as a composition of REMOVE and PUT.

**Compliance With Llm Reviewing Policy:**

Affirmed.

**Final Justification:**

Main concerns have been well addressed.

**Key Questions For Authors:**

See weaknesses

**Limitations:**

There is no discussion of limitations.

**Strengths And Weaknesses:**

**Strengths**:

1. The author use technically sound methods for understanding state tracking mechanism in LLMs, combining probes, path patching, and interventions.

2. The analysis of the REMOVE operation is particularly interesting and the most novel part of the paper.

3. Predicting new failure cases based on the REMOVE analysis further consolidates the findings.

**Weaknesses**:

1. The paper shoud have a separate related work section. Right now, the discussion of prior work is folded into the introduction, which makes it harder to clearly position this paper against the broad literature.

2. Some findings feel similar to previous work. it reuses essentially the same query-conditioned position-tracking circuit template as Prakash et al. (2024), and the conclusion that the model relies on latent positional information rather than token identity also falls closed to prior binding-ID / ordering-ID work (Feng & Steinhardt, 2023; Dai et al., 2024). At a high level, the finding that LLMs consolidate information relevant to the queried box only is similar to the main finding of Li et al. (2024a), namely that LLMs preserve only the information needed for the downstream objective rather than maintaining a general world representation.

3. The analysis is limited to a small set of operations and settings.

---

> ### Author Rebuttal · Authors · 2026-03-31
>
> Thank you for your positive review and feedback!
>
> - **Weakness 1**: We will include a separate related work section with the extra page for the camera-ready version. For example, we will include a section on probing world states (Li et al 2022; Li et al 2024a; Mamidanna et al 2025), a section on entity binding/tracking (Feng et al 2024; Li et al 2025; Prakash et al 2024; Gur-Arieh et al 2025 etc.), and a section on mechanistic interpretability methods (from probing to more recent techniques such as activation patching (Gupta et al 2015; Ravfogel 2020; Belinkov 2022 etc.)
> - **Weakness 2**: While some findings qualitatively overlap with previous work, we believe the insights we gain are distinct because they apply to a different problem domain. Previous work (Feng et al 2024; Dai et al 2024; Prakash et al 2024; Prakash et al 2025; Gur-Arieh et al 2025) only looked at entity binding, which corresponds to only the “no-op” scenarios. Our analysis is broader in scope, looking into entity tracking with state changes. Using python variable binding as an analogy, prior work looks into statements such as “x=1, y=2”, whereas we look into “x=[1],y=[2,3],x.append(4),y.remove(1)...”. The complexity of the task builds up, and behaviorally it is known that models perform worse on cases with (more) state changes than no-op cases (Kim and Schuster, 2023), which motivates the investigation. It was not known prior to our investigation that the mechanisms for state changing operations generalize from the entity binding cases until we looked into the cases. We believe it is interesting that the mechanisms do generalize, as opposed to totally different mechanisms being recruited, especially given that PUT and DESCRIPTION (~no-op) use different natural language phrasings.
> - **Weakness 3**: We believe that our settings meaningfully and substantially extend a large body of prior work that only characterizes entity binding by introducing state-changing operations. The operations we analyze may be few in number but they form the basis of more complex state changes since they can be composed/stacked to describe problems of arbitrary complexity. Furthermore, while additional mechanisms may be involved in multiple query operations (e.g. operation order resolution), one of our main findings (models do not build world state across context or local states across layers) is a generalizable takeaway that is applicable to problems with more complex state changes. If you have further reservations about the limitations of the evaluation dataset itself, please refer to our response to Reviewer J4Ka (first bullet point).
> - **Limitations**: We will make sure to add a Limitations section for the camera-ready version, covering discussions with the reviewers.
>
> ## Citations:
> - Li, K., Hopkins, A. K., Bau, D., Viégas, F., Pfister, H., & Wattenberg, M. Emergent World Representations: Exploring a Sequence Model Trained on a Synthetic Task. In The Eleventh International Conference on Learning Representations.
> - Li, B. Z., Guo, Z. C., and Andreas, J. (how) do language
> models track state? In Forty-second International Con-
> ference on Machine Learning, 2025.
> - Li, Z., Cao, Y., and Cheung, J. C. Do llms build world repre-
> sentations? probing through the lens of state abstraction.
> Advances in Neural Information Processing Systems, 37:
> 98009–98032, 2024a
> - Mamidanna, S., Rai, D., Yao, Z., & Zhou, Y. (2025, November). All for one: Llms solve mental math at the last token with information transferred from other tokens. In Proceedings of the 2025 Conference on Empirical Methods in Natural Language Processing (pp. 30735-30748).
> - Feng, J., Russell, S., and Steinhardt, J. Monitoring latent
> world states in language models with propositional probes.
> In The Thirteenth ICLR, 2024.
> - Gur-Arieh, Y., Geva, M., and Geiger, A. Mixing mech-
> anisms: How language models retrieve bound entities
> in-context.
> - Prakash, N., Shaham, T. R., Haklay, T., Belinkov, Y., and
> Bau, D. Fine-tuning enhances existing mechanisms: A
> case study on entity tracking. In The Twelfth International
> Conference on Learning Representations, 2024.
> - Prakash, N., Shapira, N., Sharma, A. S., Riedl, C., Belinkov,
> Y., Shaham, T. R., Bau, D., and Geiger, A. Language
> models use lookbacks to track beliefs.
> - Ravfogel, S., Elazar, Y., Gonen, H., Twiton, M., and Gold-
> berg, Y. Null it out: Guarding protected attributes by
> iterative nullspace projection. In Proceedings of the 58th
> Annual Meeting of the Association for Computational
> Linguistics, 2020.
> - Belinkov, Y. Probing classifiers: Promises, shortcomings,
> and advances. Computational Linguistics, 48(1):207–219,
> March 2022.
> - Gupta, A., Boleda, G., Baroni, M., and Pad´ o, S. Distribu-
> tional vectors encode referential attributes. In M`
> arquez, L., Callison-Burch, C., and Su, J. (eds.), Proceedings
> of the 2015 EMNLP
> - Kim, N. and Schuster, S. Entity tracking in language mod-
> els. In The 61st Annual Meeting Of The Association For
> Computational Linguistics, 2023.

---

> > ### Author Rebuttal · Reviewer_BhY1 · 2026-04-04
> >
> > Thank you for your response. I still learn toward accepting the paper.

---

### Decision · Program_Chairs · 2026-04-30

**Decision:**

Accept (regular)

**Comment:**

This paper studies the entity tracking mechanism of LLMs. Specifically, they study how LLM track entities gone through state-changing oeprations such as PUT, REMOVE, MOVE, e.g.,

"The apple is in Box 0, the peach is in Box 1, the
clock and the jar is in Box 2, the television is in Box
3, the brain is in Box 4, the book is in Box 5, the pin
is in Box 7. Put the watch in Box 1. Remove the jar
from Box 2. Move the apple from Box 0 to Box 1.
Box 1 contains the"

An interesting result they found through mechanistic intepretability tools such as probing and path analysis is: LLMs do not build world states incrementally across tokens or layers, but immediately aggregate relevant information ath the last token when the query becomes clear in a pattern matching way, which seems different from the human way. All reviewers gave positive scores and confirmed the solid contribution to new understanding of LLMs' entity tracking mechanism.